# Modelling sequential branching dynamics with a multivariate branching Gaussian process

**Elvijs Sarkans**                                                                 *elvijs.sarkans@gmail.com*
*BIOS Health*

**Sumon Ahmed**                                                                          *sumon@du.ac.bd*
*University of Dhaka*

**Magnus Rattray**                                                         *magnus.rattray@manchester.ac.uk*
*University of Manchester*

**Alexis Boukouvalas**                                                     *alexis.boukouvalas@gmail.com*
*PROWLER.io*

**Reviewed on OpenReview:** *https://openreview.net/forum?id=9KoBOlstTq*

## Abstract

The Branching Gaussian Process (BGP) model is a modification of the Overlapping Mixture of Gaussian Processes (OMGP) where latent functions branch in time. The BGP model was introduced as a method to model bifurcations in single-cell gene expression data and order genes by inferring their branching time parameter. A limitation of the current BGP model is that the assignment of observations to latent functions is inferred independently for each output dimension (gene). This leads to inconsistent assignments across outputs and reduces the accuracy of branching time inference. Here, we propose a multivariate branching Gaussian process (MBGP) model to perform joint branch assignment inference across multiple output dimensions. This ensures that branch assignments are consistent and leverages more data for branching time inference. Model inference is more challenging than for the original BGP or OMGP models because assignment labels can switch from trunk to branch lineages as branching times change during inference. To scale up inference to large datasets we use sparse variational Bayesian inference. We examine the effectiveness of our approach on synthetic data and a single-cell RNA-Seq dataset from mouse haematopoietic stem cells (HSCs). Our approach ensures assignment consistency by design and achieves improved accuracy in branching time inference and assignment accuracy.

## 1 Introduction

Many algorithms have been developed to uncover trajectories from single-cell gene expression data (Saelens et al., 2019). Genome-scale single-cell experiments are destructive, so that time information is lost, and trajectory inference provides a useful approach to associate cells with inferred time labels (pseudotimes) to help recover gene expression dynamics. The resulting trajectories are of great biological interest and branching trajectories are of particular interest in systems where cells can differentiate into multiple alternative fates.

Pseudotime inference methods capture broad changes across high-dimensional gene expression data. For example, a popular approach is the Monocle algorithm (Qiu et al., 2017), which combines dimensionality reduction with minimal spanning tree estimation to order cells along trajectories. However, these algorithms do not model differences in branching dynamics across individual genes. In this paper, we consider the complementary problem of modelling gene-specific branching dynamics along pseudotime trajectories. This is biologically important as it identifies the branching order of different genes, leading to a better understanding of gene networks involved in cellular differentiation. Two existing methods tackle the problem of

inferring gene-specific branching times: Branch Expression Analysis Modelling (BEAM) (Qiu et al., 2017) and Branching Gaussian Process (BGP) (Boukouvalas et al., 2018). BEAM was the first method to tackle the branching time inference problem by using a relatively straightforward spline-based approach. Boukouvalas et al. (2018) improved on the branching time estimation with BGP, especially for early branching genes and genes with low signal-to-noise. They achieved this improved performance by combining trajectory inference with branch label inference to better identify gene-specific branching times. The PseudotimeDE (Song & Li, 2021) and Tradeseq (Van den Berge et al., 2020) methods have also been proposed to estimate which genes are branching. However, these methods do not infer the branching time and cannot be used in ordering the branching times of different genes.

BGP uses an approach inspired by the Overlapping Mixture of Gaussian Processes model (OMGP, Lázaro-Gredilla et al., 2012) to probabilistically label cells with the branches they belong to. OMGP is a mixture model developed for time-series data where each mixture component is a Gaussian Process (GP) function and data points can be assigned to alternative mixture components. In the case of single-cell gene expression data, pseudotimes represent cells' progression through some developmental process and are analogous to time-series labels. Therefore, each cell can be assigned to the mixture components of OMGP based on pseudotime, an approach adopted by the GPfates method for trajectory inference (Lönnberg et al., 2017). However, in the standard OMGP, mixture components are independent and are represented by *a priori* independent latent GP functions, whereas cellular trajectories are typically branching as cells begin in the same state but later differentiate into one of multiple alternative states during a developmental process. Therefore, Boukouvalas et al. (2018) adapted the OMGP model by introducing dependence among the mixture components. Their BGP model uses a branching kernel that forces latent GP functions representing mixture components to intersect at a branching location and hence to become dependent on one another. Before the branching time cells are assigned to a trunk function and after the branching time cells can be assigned to either of two child functions. The branching time is a parameter of the model that can be inferred to distinguish early, late and non-branching genes.

Trajectory inference methods such as Monocle can be used to provide cellular branch label information after some global cellular branching time inferred from the reduced dimension gene expression data. However, early-branching genes may be incorrectly assigned to the trunk according to this global assignment. The BGP model allows the branching time for specific genes to be earlier than the global branching time and new branch labels to be inferred after this time. Boukouvalas et al. (2018) showed the BGP approach outperformed BEAM, especially with regards to such early branching genes for which no global branching information is available. However, in the original BGP model the branch assignments were made independently for each gene and could potentially be inconsistent across genes, which makes little biological sense. Here, we extend the model to address this limitation. In the proposed multivariate BGP (MBGP) we perform joint inference across all genes in a way that ensures consistency in cell assignments. This increases the power of the method to infer branch labels, so that MBGP is better able to correct errors in the Monocle branch labelling and can assign cells to branches with much weaker prior label information than BGP. An added benefit is that a single fit of the model on many genes (MBGP) is faster than many fits on a single gene (BGP) due to the label inference being shared.

In Section 2 we define the modelling problem. In Section 3 we demonstrate the limitations of the BGP model on single cell gene expression data. In Section 4 we describe the MBGP model and in Section 5 we compare its performance to the original BGP model on both synthetic and single-cell gene expression data.

## 2 Problem definition

Consider observations $\boldsymbol{Y} \in \mathbb{R}^{N \times D}$ associated with inputs $\boldsymbol{t} \in \mathbb{R}^N$ (without loss of generality, assumed to be monotonically increasing and in $[0, 1]$) with the following properties:

- Inputs $\boldsymbol{t}$ define a branching process for each dimension $d$. Prior to an unknown branching point $b_d \in [0, 1]$, i.e. when $t_n < b_d$, data point $\boldsymbol{Y}_{n,d}$ is associated with a "trunk" while after this point data points can be associated with one of two underlying bifurcation "branches" that generated it.

- The minimum branching point across dimensions defines the global branching point. After this point each input $t_n$ is associated with one of two global bifurcation branches.

- The assignment of data points to branches is consistent with the global branching across outputs (but unknown). That is, there exists a global assignment of inputs $t_n$ to branches, such that if $t_n$ is in branch A, then all outputs $Y_{n,d}$ with $t_n > b_d$ are also associated with branch A.

For datasets that satisfy the properties above, we want to solve two problems:

- **branching time inference**: for each output $d \in \{1, 2, ..., D\}$ determine the branching time $b_d$.

- **data point labeling**: for each data point $n \in \{1, 2, ..., N\}$ determine which branch it belongs to.

The formal definition is somewhat dense, but it tries to capture a simple idea illustrated in Figure 1.

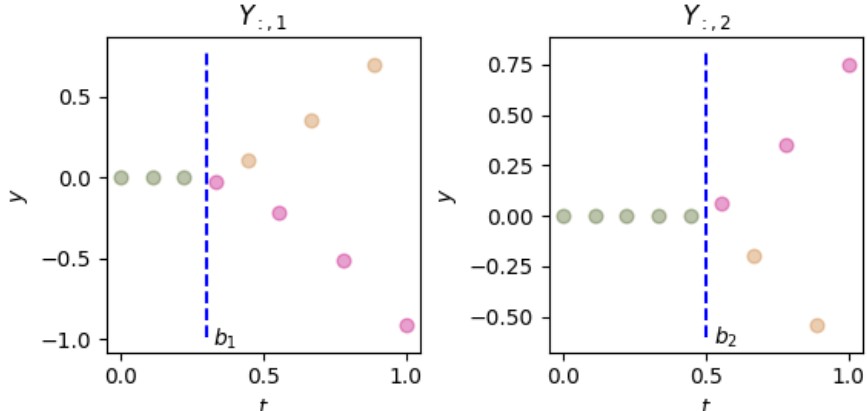

Figure 1: Illustration of the problem definition. We plot a dataset with $D = 2$ outputs over $N = 10$ data points corresponding to equidistant inputs on $[0, 1]$: $t_1 = 0, t_2 = 0.11..., ..., t_{10} = 1.0$. The branching time inference problem is solved by $b_1 = 0.3$ and $b_2 = 0.5$. The data point labeling problem is solved via data point assignments indicated by the data point colours. The trunk is coloured green while the two branches (A and B) are coloured yellow and pink respectively. In a biological context, the data $Y_{:,1}$ corresponds to a gene that branches early and whose expression rises in branch A and falls in branch B, while the data $Y_{:,2}$ corresponds to a gene that branches later, whose expression falls in branch A and rises in branch B. The cell assignments are consistent once both genes have started to branch.

As discussed in the introduction, the abstract problem definition is motivated by the trajectory inference problem in single-cell gene expression data. In this application, $Y_{n,d}$ measures how much cell $n$ expresses gene $d$ at input $t_n$ (usually called "pseudotime"). The branching points represent when the expression of a gene begins to differ between two lineages (branches). The "consistent" data point assignment to branches captures the idea that different cells evolve via consistent trajectories.

We note that no "ground truth" branching points or cell labels are available for real datasets, which makes the algorithm evaluation somewhat subjective. We will therefore augment the evaluation of MBGP to include synthetic datasets with known ground truth.

## 3 Branching Gaussian Process (BGP): Overview and limitations

We highlight some of the key equations and techniques used in the original BGP method (Boukouvalas et al., 2018) that we will be building on in §4. We show how it is a good model for solving the branching time inference, but makes errors in the data point labeling problem and does not ensure data point labeling consistency across outputs.

The outputs in BGP are modelled as independent. This makes the extension from one to many outputs straightforward (see §3.3 for details). However, as we will see in §3.4 this mathematical convenience comes at a cost to modelling capabilities.

## 3.1 Model specification

Let $\boldsymbol{y} \in \mathbb{R}^{N \times 1}$ be single output data conforming to the full problem definition from §2. We use latent GP functions to represent the trunk and branches of the data.

We first discuss the kernel construction that captures the requirement that the trunk and branch latent functions cross in a single location - the branching or bifurcation point. We then write down the model likelihood, and finally we briefly discuss the inference.

### 3.1.1 Branching kernel

For clarity we restrict our attention to three latent GP functions $\{f_1, f_2, f_3\}$ that represent the trunk and branches of the bifurcation. We will sometimes also denote these three latent functions by $\{f, g, h\}$ with $f$ denoting the trunk. For convenience, an evaluation of latent functions at $N$ inputs (corresponding to cells in $\boldsymbol{y}$) is denoted by $\boldsymbol{F^c} \in \mathbb{R}^{3 \times N}$ where $f_i$ corresponds to the $i$-th row ($1 \leq i \leq 3$).

Notation: we use $\boldsymbol{F^c}$ to describe the latent functions using OMGP-compatible matrix shapes. In Section §3.3 we will define a reshaped but equivalent $\boldsymbol{F}$ that makes the variational inference equations simpler.

In order for the latent trunk and branch functions to capture bifurcations, we need to constrain them all to cross at a given input point. Yang et al. (2016) described the joint probability density and the corresponding GP covariance for two latent functions constrained to cross each other at a given input. This was extended by Boukouvalas et al. (2018) to three latent functions crossing at a single point. We briefly re-state the key governing equations.

We place the GP priors on $f$, $g$, $h$, constrain them to all cross at the branching time $b$, and restrict the prior covariances to all have the same structure $k$:

$$\begin{aligned} f(t) &\sim \mathcal{GP}(0, k(t, t')), \\ g(t) &\sim \mathcal{GP}(0, k(t, t')), \\ h(t) &\sim \mathcal{GP}(0, k(t, t')), \\ f(b) &= g(b) = h(b). \end{aligned}$$

Let us consider a vector of times $\boldsymbol{t}$ and evaluate the covariance of the random variables $f(\boldsymbol{t})$, $g(\boldsymbol{t})$, $h(\boldsymbol{t})$, denoted by $\boldsymbol{S}$. Since the individual covariance structures of $f$, $g$ and $h$ are identical, the diagonal blocks of $\boldsymbol{S}$ are given by $\boldsymbol{K}_{i,j} := k(\boldsymbol{t}_i, \boldsymbol{t}_j)$, whereas off-diagonal blocks are given by

$$\boldsymbol{L}_{i,j} := \frac{k(\boldsymbol{t}_i, b)k(b, \boldsymbol{t}_j)}{k(b, b)},$$

giving us the overall covariance

$$\boldsymbol{S} = \begin{pmatrix} \boldsymbol{K} & \boldsymbol{L} & \boldsymbol{L} \\ \boldsymbol{L} & \boldsymbol{K} & \boldsymbol{L} \\ \boldsymbol{L} & \boldsymbol{L} & \boldsymbol{K} \end{pmatrix}. \tag{1}$$

Figure 2 shows an example of this covariance structure. The diagonal blocks show the covariance structure of kernel $k$ (in this case, an exponentiated quadratic) whereas the off-diagonal blocks reflect the intersection of latent functions at time $b$ where the functions are constrained to be equal.

Note that the branching kernel can be extended to capture multiple bifurcations by extending the number of latent functions considered. This general case can be treated in much the same way as the single bifurcation case. However, here we restrict our attention to modelling a single bifurcation event.

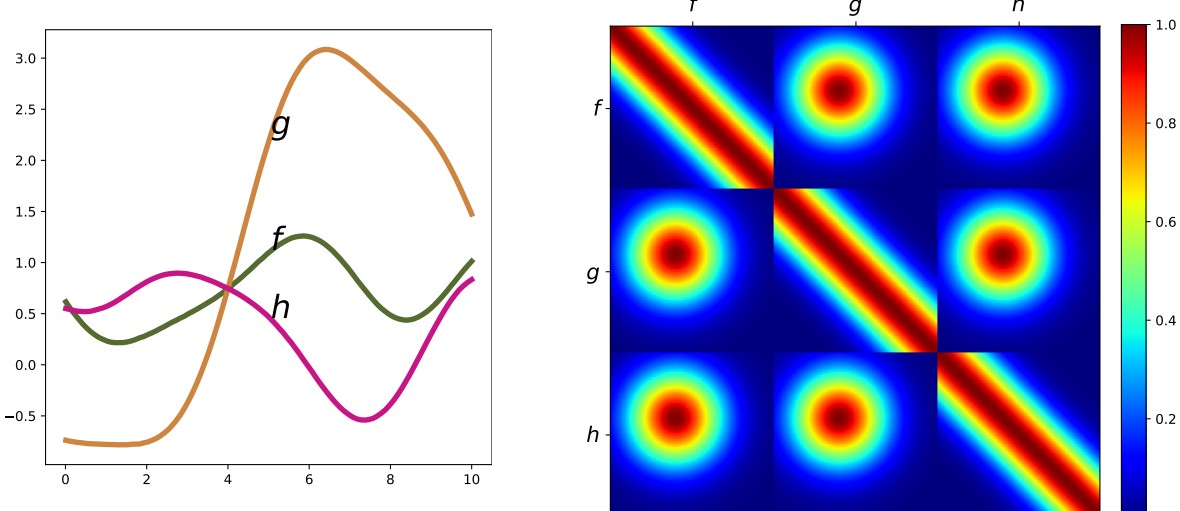

Figure 2: Illustration of the covariance matrices used by Boukouvalas et al. (2018). The input $\boldsymbol{t}$ contains 100 evenly distributed time points within the range $[0, 10]$ for each latent function, and the global branching time is set to $b = 4$. For the joint covariance structure $k$ we have used an exponentiated quadratic. Three latent functions $f, g$ and $h$ are intersecting at the branching time $b$ (left sub-panel) and the covariance matrix from which these three functions have been sampled (right sub-panel). The covariance matrix is evaluated at every point within the input domain $[0, 10]$ for all three functions.

### 3.1.2  Likelihood

Let $\boldsymbol{Z^c} \in \{0, 1\}^{N \times 3}$ be a binary indicator matrix determining the assignment of $N$ data points to precisely one of the three latent functions. That is, each row of $\boldsymbol{Z^c}$ has precisely one nonzero entry. The model likelihood is given by

$$p(\boldsymbol{y}|\boldsymbol{F^c}, \boldsymbol{Z^c}) := \mathcal{N}(\boldsymbol{y}; \operatorname{diag}(\boldsymbol{Z^c F^c}), \sigma^2 \boldsymbol{I}_n). \tag{2}$$

We note that the extension to multiple outputs is done by employing a different $\boldsymbol{Z^c}$ for each gene, all of which are independent. This makes the likelihood factorise, so that the extension is mathematically straightforward and we omit its discussion from this section.

As in Lázaro-Gredilla et al. (2012), a categorical prior was placed on the indicator matrix $\boldsymbol{Z^c}$:

$$P(\boldsymbol{Z^c}) = \prod_{n=1}^{N} \prod_{i=1}^{3} {H^c}_{n,i}^{{Z^c}_{n,i}}, \tag{3}$$

where $\boldsymbol{H^c} \in \mathbb{R}^{N \times 3}$ is the data point assignment probability matrix. That is, ${H^c}_{n,i}$ represents the probability of cell $n$ being assigned to latent function $i$, so that $\sum_{i=1}^{3} {H^c}_{n,i} = 1$ for all $n$. We note that in the many-outputs case there is a different $\boldsymbol{Z^c}$ and $\boldsymbol{H^c}$ for each gene.

### 3.2  Equivalence with OMGP

Whilst BGP was inspired by OMGP, the use of somewhat different notation means that the model likelihood equivalence is not obvious. Let us show it explicitly.

Equation (4) from OMGP (Lázaro-Gredilla et al., 2012) defines $p(\boldsymbol{y}|\boldsymbol{F^c}, \boldsymbol{Z^c}) := \prod_{n=1}^{N} \prod_{i=1}^{3} \mathcal{N}(y_n; F^c_{i,n}, \sigma^2)^{Z^c_{n,i}}$. Recall that $\boldsymbol{Z^c}$ has precisely one nonzero entry in each row. We

denote its index for row $n$ to be $i_n$, so that $Z^c_{n,i_n} = 1$. Then

$$\prod_{i=n}^{N}\prod_{i=1}^{3}\mathcal{N}(y_n; F^c_{i,n}, \sigma^2)^{Z^c_{n,i}} = \prod_{n=1}^{N}\mathcal{N}(y_n; F^c_{i_n,n}, \sigma^2) = \prod_{n=1}^{N}\mathcal{N}\left(y_n; \sum_{i=1}^{3}Z^c_{n,i}F^c_{i,n}, \sigma^2\right) = \mathcal{N}(\boldsymbol{y}; \mathrm{diag}(\boldsymbol{Z^c F^c}), \sigma^2)$$

(4)

thus establishing our claim. Note that we are considering only three branches, but the equivalence would hold for more general branching kernels as well since the derivation above does not depend on the number of branches.

### 3.3 Inference

The BGP inference scheme is based on an important re-statement of the model likelihood in (2). Let the rows of $\boldsymbol{F^c}$ be given by $\{\boldsymbol{f}, \boldsymbol{g}, \boldsymbol{h}\}$. Define a reshaped $\boldsymbol{F} := (f_1, g_1, h_1, f_2, g_2, h_2, \ldots, f_N, g_N, h_N)^T \in \mathbb{R}^{3N\times 1}$. Given $\boldsymbol{Z^c}$ as above with $Z^c_{n,i_n} = 1$ (as in §3.2), define $\boldsymbol{Z} \in \{0,1\}^{N\times 3N}$ to be a binary indicator matrix with 0 everywhere except for one element on each row: $Z_{n,3(n-1)+i_n} = 1$ for $n = 1, 2, \ldots, N$.

We highlight that the important consequence of the above definitions is that $\mathrm{diag}(\boldsymbol{Z^c F^c}) = \boldsymbol{ZF}$ since

$$\mathrm{diag}(\boldsymbol{Z^c F^c})_n = \sum_{i=1}^{3}Z^c_{n,i}F^c_{i,n} = F^c_{i_n,n} = F_{3(n-1)+i_n} = \sum_{i=1}^{3N}Z_{n,i}F_i = (\boldsymbol{ZF})_n.$$

This gives us an alternative formulation of (2):

$$p(\boldsymbol{y}|\boldsymbol{F}, \boldsymbol{Z}) := p(\boldsymbol{y}|\boldsymbol{F^c}, \boldsymbol{Z^c}) = \mathcal{N}(\boldsymbol{y}; \mathrm{diag}(\boldsymbol{Z^c F^c}), \sigma^2\boldsymbol{I}_n) = \mathcal{N}(\boldsymbol{y}; \boldsymbol{ZF}, \sigma^2\boldsymbol{I}_n). \tag{5}$$

The log-likelihood of the model is not analytically tractable as it requires integrating out the indicator matrix $\boldsymbol{Z}$. Boukouvalas et al. (2018) used variational inference to derive an analytical solution for the lower bound as in Lázaro-Gredilla et al. (2012). More precisely, they modelled the distributions of branch assignment $\boldsymbol{Z}$ and latent processes $\boldsymbol{F}$ as independent: $q(\boldsymbol{Z}, \boldsymbol{F}) = Q(\boldsymbol{Z})q(\boldsymbol{F})$; this is sometimes referred to as the mean-field approximation. A standard variational lower bound on the marginal likelihood is derived using Jensen's inequality (see e.g. King & Lawrence, 2006)

$$\log p(\boldsymbol{y}|\boldsymbol{F}) \geq \mathbb{E}_{Q(\boldsymbol{Z})}[\log p(\boldsymbol{y}|\boldsymbol{F}, \boldsymbol{Z})] - D_{\mathrm{KL}}[Q(\boldsymbol{Z})\|P(\boldsymbol{Z})]. \tag{6}$$

Boukouvalas et al. (2018) then chose to model data point assignments to latent branches $Z$ as independent Bernoulli-like variables for each data point. That is, $Q(\boldsymbol{Z}) := \prod_{n=1}^{N}\prod_{i=1}^{3N}U_{n,i}^{Z_{n,i}}$ where $U_{n,i}$ is the posterior probability of data point $n$ being assigned to a particular branch $i_n = 1, 2, 3$ with $i = 3(n-1)+i_n$ (probability is 0 for all other outcomes). The branch can be either the trunk state when $t_n \leq b$, or one of the branches after bifurcation when $t_n > b$. By denoting $\boldsymbol{U} := \mathbb{E}_{Q(\boldsymbol{Z})}[\boldsymbol{Z}]$ they arrived at an important bound on the KL divergence for the above choice of $Q(\boldsymbol{Z})$ (see Equations (18)-(19) from Boukouvalas et al. (2018))

$$D_{\mathrm{KL}}[Q(\boldsymbol{Z})\|P(\boldsymbol{Z})] = \sum_{n=1}^{N}\sum_{i=1}^{3N}U_{n,i}\log\left(\frac{U_{n,i}}{H_{n,i}}\right). \tag{7}$$

where we have defined $\boldsymbol{H}$ as a counterpart to $\boldsymbol{H^c}$ from (3) in the obvious way. Next, $\boldsymbol{F}$ is integrated out to derive an exact variational collapsed bound of the marginal log-likelihood $\log p(\boldsymbol{y})$ (Boukouvalas et al., 2018). The model's parameters can be estimated by maximising a variational bound on the log-likelihood.

Boukouvalas et al. (2018) also derived a sparse approximation bound of their model using inducing points that allows the BGP model to scale up to larger datasets.

An important aspect of the BGP is that the branching time posterior probability is not available analytically and is calculated numerically using the approximate marginal likelihood evaluated at a uniformly spaced set of candidate branching points. This approach would scale exponentially with the number of branching times in the MBGP model that we introduce below and therefore we use an alternative gradient computation to estimate only the mode of the branching posterior.

### 3.4 Limitations of the method

BGP is a good model for solving the branching time inference problem as evidenced by comparisons with the relevant methods, i.e. BEAM (Qiu et al., 2017) and the Mixture of Factor Analysers (Campbell & Yau, 2017), see Boukouvalas et al. (2018).

Having said that, its applicability is hindered by its shortcomings in solving the data point labelling problem. The BGP model assigns data points to branches output-by-output without ensuring consistency across outputs (see section §2). That is, a data point may be assigned to branch A for one output and to branch B for another output.

Notably, when applied to biological datasets analysing early branching genes that branch before the global branching time identified by a pseudotime algorithm, then no prior information on cell branch assignment prior to the global branching time is available. In this case it is likely to be beneficial to leverage data from multiple genes to consistently solve the cell labelling problem, since there will be high uncertainty given data from only one gene. We note also that the gene-by-gene inference is computationally inefficient.

To explore these issues empirically, have applied the BGP method on a single cell mouse hematopoietic stem cell dataset (Paul et al., 2015). The data contain 4423 cells and a global pseudotime and branching structure was inferred using Monocle 2 (Qiu et al., 2016). Details of the experimental setup are given in the supplementary material.

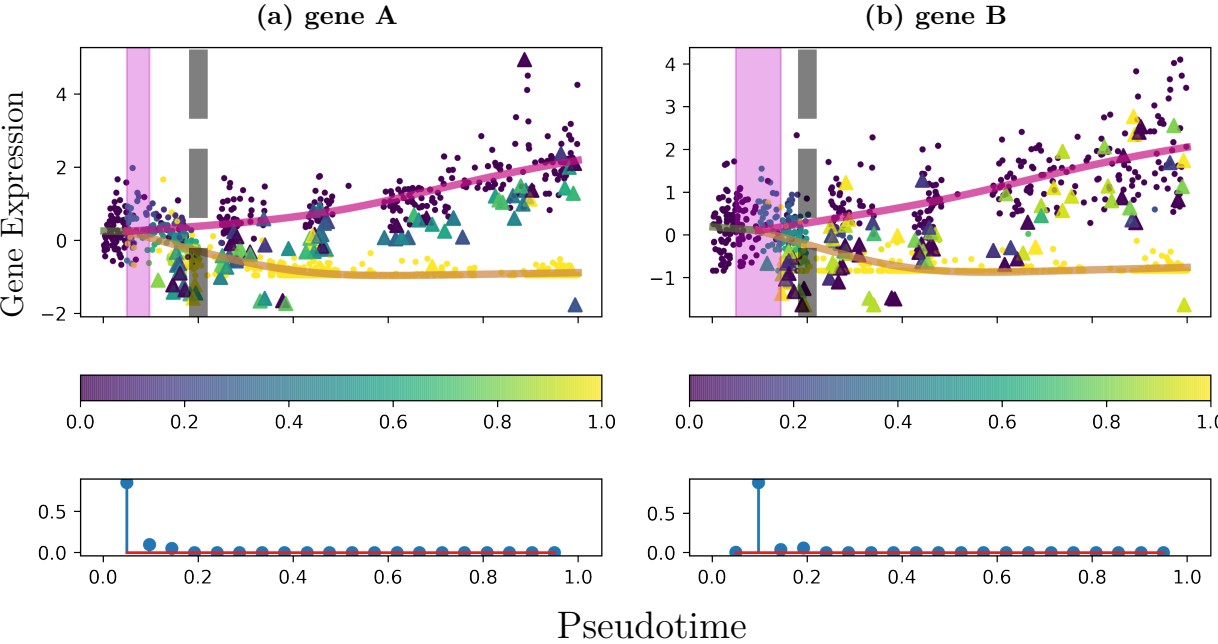

Figure 3: Mouse haematopoietic stem cells (Paul et al., 2015): Inconsistent cell assignment between the pair of early branching genes MPO (gene A) and CTSG (gene B). The 1000 randomly sub-sampled cells that have been used for inference are coloured according to the posterior global branching pattern derived using the BGP algorithm (yellow means more likely to be in branch A, blue means more likely to be in branch B). The points are used to represent the cells have been assigned to the same branches, i.e. consistent across these two genes. On the other hand, the triangles represent the inconsistent cell assignment, i.e. cells that have been assigned to different branches across these two genes.

Boukouvalas et al. (2018) tried to mitigate the risk of inconsistent cell assignments by placing a strong global prior over the cell assignments, but the problem can remain even when a strong prior is used. For instance, Figure 3 shows the posterior cell assignment of randomly sub-sampled 1000 cells for two early branching genes MPO and CTSG. To avoid the inconsistent cell assignment a very strong prior confidence 0.95 was

used in both cases. This strong prior assumes that cells after the global branching point in pseudotime ordering are unlikely to switch their branching assignment away from the prior. Among the 1000 cells that are used in the inference, we have found that 88 cells (triangle markers) have inconsistent cell assignments, i.e. have been assigned to different branches in the posterior. The posterior cell assignment inconsistency on the eight biomarkers of HSCs that show very strong evidence of branching has been summarised in Table 1. Posterior cell assignments have been calculated using a high confidence (0.80) for all eight genes. For each gene, different numbers of cells show strong evidence of being assigned to the different branches. The cells may actually belong to a particular branch or they have become biased towards that branch in case of the gene under consideration. For each pair of genes, we identify the number of cells that have been assigned to different branches in posterior by each gene. We find that among the 1000 cells used in the inference of BGP, 468 cells show inconsistency across all eight genes.

Table 1: Number of cells (among 1000 cells used in the inference) inconsistently assigned to different branches by BGP for each pair of the eight biomarkers of hematopoietic stem cells (HSCs).

|        | MPO | PRTN3 | CALR | CAR1 | CAR2 | GSTM1 | ELANE |
|--------|-----|-------|------|------|------|-------|-------|
| CTSG   | 88  | 99    | 75   | 230  | 156  | 158   | 76    |
| MPO    |     | 81    | 127  | 288  | 201  | 213   | 115   |
| PRTN3  |     |       | 147  | 299  | 219  | 226   | 136   |
| CALR   |     |       |      | 223  | 146  | 147   | 72    |
| CAR1   |     |       |      |      | 191  | 205   | 182   |
| CAR2   |     |       |      |      |      | 159   | 94    |
| GSTM1  |     |       |      |      |      |       | 105   |

The second limitation of the BGP model is computational complexity. Since BGP estimates the branching time posterior by using the normalised approximated likelihood calculated at a set of candidate branching points over the input domain, the computational requirements are high when analysing many outputs. Although the model inference can be done in parallel (per output), this still requires significant compute. In our experiments reported in Section 5, BGP on a 1000 data point dataset takes around one hour per output for a total of 8 hours across all 8 outputs whilst MBGP takes around 30 mins for all outputs[1]. Boukouvalas et al. (2018) had to sub-sample the data points in order to minimise the per-output computational time, but the computational requirements still remain very high. This exceptionally high computational load makes the BGP model impractical in many instances.

## 4 Multivariate Branching Gaussian Process (MBGP)

We propose a Multivariate Branching Gaussian Process (MBGP) model that performs inference jointly across outputs. This approach addresses BGP's data point label inconsistency issue, is computationally faster and improves the statistical power available to infer branch assignments by jointly considering all outputs.

The MBGP model builds on the BGP model discussed in §3.1. It uses the same branching kernel (see §3.1.1). It also uses the same likelihood from (5), but with one modification – the $Z$ and $H$ assignment matrices and prior probabilities are shared across all outputs. This means the model no longer factorises across outputs and hence we develop a new variational approximation for the marginal likelihood. This change to the model addresses the consistent data point labelling problem from §2 by ensuring consistent data point labelings across all outputs.

Another change in MBGP is its treatment of branching times. Recall that §3.3 described how the branching times were obtained by comparing approximate marginal likelihoods on a grid of candidate points. As this does not scale to many outputs due to the curse of dimensionality, we treat branching times as model parameters that are optimised along with other parameters.

---

[1]System configuration: Intel(R) Core(TM) i7-1065G7 CPU @ 1.30GHz 1.50 GHz, 16 GB RAM, 64-bit operating system, x64-based processor.

### 4.1 Model and variational bound

The likelihood factorises for multiple independent outputs so that the extension of the single-output likelihood in (5) to multiple independent outputs is straightforward

$$p\left(\boldsymbol{Y}|\boldsymbol{F},\boldsymbol{Z}\right) = \prod_{d=1}^{D} \mathcal{N}\left(\boldsymbol{y}_d; \boldsymbol{Z}\boldsymbol{F}_d, \sigma^2 \mathbf{I}\right), \tag{8}$$

where $D$ is the number of outputs, $\boldsymbol{y}_d$ is the $d^{th}$ column of the data matrix $\boldsymbol{Y} = \{\boldsymbol{y}_1, ..., \boldsymbol{y}_D\}$ and $\boldsymbol{F}_d$ denotes the $d^{th}$ column vector from the set of latent GP functions $\boldsymbol{F} = \{\boldsymbol{F}_1, ..., \boldsymbol{F}_D\}$ .

Crucially, the association matrix $\boldsymbol{Z}$ is shared by all outputs with a categorical prior (equation 3) placed over $\boldsymbol{Z}$. We treat the branching time for each output $b_d$ as an explicit parameter of this prior. The prior over the latent functions $\boldsymbol{F}_d$ for each output is a branching GP also parameterised by the branching time and kernel parameters $\theta_d$,

$$P\left(\boldsymbol{Z}|b_d\right) = \prod_{n=1}^{N}\prod_{i=1}^{3N} H_{n,i}(b_d)^{Z_{n,i}} ,$$

$$\boldsymbol{F}_d|b_d \sim \mathcal{GP}\left(0, k(\theta_d)|b_d\right) \text{ for } d = 1, 2, \dots, D .$$

The categorical prior is adjusted in an output-specific manner so that data points are either assigned to the trunk or the two branches depending on the branching time. That is, $H(b)_{n,3(n-1)+1} = 1$ if $t_n < b$, while $[H(b)_{n,3(n-1)+2}, H(b)_{n,3(n-1)+3}] = [h_n, 1 - h_n]$ otherwise, with all other entries in row $n$ set to zero. The variational approximation for the assignment probabilities $Q(\boldsymbol{Z})$ has the same structure

$$Q\left(\boldsymbol{Z}|b_d\right) = \prod_{n=1}^{N}\prod_{i=1}^{3N} U_{n,i}(b_d)^{Z_{n,i}}, \tag{9}$$

with $U(b)_{n,3(n-1)+1} = 1$ if $t_n < b$, while $[U(b)_{n,3(n-1)+2}, U(b)_{n,3(n-1)+3}] = [u_n, 1 - u_n]$ otherwise, where $u_n$ is the inferred probability that data point $n$ is assigned to the first branch.

The log likelihood terms also depend on the branching parameter. If we denote $\boldsymbol{b} \triangleq (b_1, ...b_D) \in \mathbb{R}^D$ and integrate over $Q(\boldsymbol{Z}|\boldsymbol{b})$, then - by using Jensen's inequality in the standard way to bring log inside the integral - we have

$$\log p\left(\boldsymbol{Y}|\boldsymbol{F},\boldsymbol{b}\right) \geq \sum_{d=1}^{D}\left[\mathbb{E}_{Q(\boldsymbol{Z}|b_d)}\left[\log p\left(\boldsymbol{y}_d|\boldsymbol{F}_d,\boldsymbol{Z},b_d\right)\right]\right] - \sum_{d=1}^{D} D_{\mathrm{KL}}\left[Q\left(\boldsymbol{Z}|b_d\right)||P\left(\boldsymbol{Z}|b_d\right)\right]$$

$$\geq -\frac{ND}{2}\log(2\pi\sigma^2) - \sum_{d=1}^{D}\frac{1}{2\sigma^2}\left(\boldsymbol{y}_d^\top \boldsymbol{y}_d + \boldsymbol{F}_d^\top \boldsymbol{A}_d \boldsymbol{F}_d - 2\boldsymbol{F}_d^\top \boldsymbol{U}_d^\top \boldsymbol{y}_d\right)$$

$$- \sum_{d=1}^{D} D_{\mathrm{KL}}\left[Q\left(\boldsymbol{Z}|b_d\right)||P\left(\boldsymbol{Z}|b_d\right)\right], \tag{10}$$

where $\boldsymbol{U}_d \triangleq \mathbb{E}_{Q(\boldsymbol{Z}|b_d)}\left(\boldsymbol{Z}\right)$ and $\boldsymbol{A}_d \triangleq \mathbb{E}_{Q(\boldsymbol{Z}|b_d)}\left(\boldsymbol{Z}^T\boldsymbol{Z}\right)$ are the variational expectations conditioned on the output-specific branching time $b_d$.

We note that $\boldsymbol{U}_d$ is straightforward to compute given equation 9, but $\boldsymbol{A}_d$ is not. Computing it explicitly is computationally prohibitive, so we establish a key identity.

$$(\boldsymbol{A}_d)_{i,j} = \mathbb{E}_{Q(\boldsymbol{Z}|b_d)}\left[(Z^T Z)_{i,j}\right] = \mathbb{E}_{Q(\boldsymbol{Z}|b_d)}\left[\sum_{k=1}^{N} Z_{k,i} Z_{k,j}\right]$$

$$(\text{separate out the } i=j \text{ case}) = \mathbb{E}_{Q(\boldsymbol{Z}|b_d)}\left[\sum_{k=1}^{N} Z_{k,i} Z_{k,j}\right](1-\delta_{ij}) + \mathbb{E}_{Q(\boldsymbol{Z}|b_d)}\left[\sum_{k=1}^{N} Z_{k,i} Z_{k,i}\right]\delta_{ij}$$

$$(Z_{k,i}^2 = Z_{k,i} \text{ as it's binary}) = \mathbb{E}_{Q(\boldsymbol{Z}|b_d)}\left[\sum_{k=1}^{N} Z_{k,i} Z_{k,j}\right](1-\delta_{ij}) + \mathbb{E}_{Q(\boldsymbol{Z}|b_d)}\left[\sum_{k=1}^{N} Z_{k,i}\right]\delta_{ij}$$

$$(Z_{k,i} Z_{k,j} = 0 \text{ when } i \neq j) = \mathbb{E}_{Q(\boldsymbol{Z}|b_d)}\left[\sum_{k=1}^{N} Z_{k,i}\right]\delta_{ij}. \tag{11}$$

Therefore, $\boldsymbol{A}_d$ is diagonal and the individual terms are now easy to compute using equation 9 and equation 11.

Finally, we note that a counterpart of equation 7 holds via a straightforward computation

$$D_{\mathrm{KL}}\left[Q(\boldsymbol{Z}|b_d)\|P(\boldsymbol{Z}|b_d)\right] = \sum_{n=1}^{N}\sum_{i=1}^{3N} U_{n,i}(b_d)\log\left(\frac{U_{n,i}(b_d)}{H_{n,i}(b_d)}\right). \tag{12}$$

By plugging equation 12 and equation 11 into equation 10, we finally obtain a computationally feasible variational lower bound on the likelihood $\log p\left(\boldsymbol{Y}|\boldsymbol{F},\boldsymbol{b}\right)$. Finally, we obtain a sparse collapsed variational bound by integrating out the latent functions $\boldsymbol{F}$ and introducing an inducing point approximation. This is relatively straightforward and well-known, so we only provide the full details in the appendix.

### 4.2 Inference

Although it is possible to estimate the lengthscale and process variance parameters jointly with the branching points, we prefer to estimate them prior to inferring branching locations. We set the branching location of every output at the starting of time (i.e. $b \rightarrow 0$) and estimate the lengthscale and the process variance parameters for each output by maximum likelihood. After estimating lengthscale and process variance, the only remaining parameters to estimate are the branching time vector and data point assignments ($\boldsymbol{Z}$). Using this approach we assume that the bifurcating functions have similar smoothness and variance before and after the branching time. This allows us to simplify inference by reducing the number of effective parameters that need to be estimated. We also employ multiple restarts to reduce the effect of local minima.

By doing gradient search we no longer infer the posterior over branching times as in the original BGP model. If a full Bayesian posterior over branching times is required then an MCMC approach can be adopted, although that would be computationally demanding.

## 5 Experimental results

We demonstrate our approach by fitting both MBGP and BGP models to noisy synthetic data. By fitting models to realistic noisy samples, we will see that MBGP substantially improves the data point assignment and improves the branching time learning when compared to BGP. We will also fit the MBGP model to real data, namely, single-cell RNA-seq data from mouse hematopoietic stem cells (Paul et al., 2015) and see how it resolves the cell label inconsistency problem whilst retaining biologically sensible fits.

### 5.1 Synthetic noisy data

In many real-world applications, there is no ground truth for data point assignments and branching times. This can make it difficult to compare methods in a quantitative manner. In order to address this, we will draw samples from a fixed MBGP model where both the ground truth branching points as well as data point assignments are known. This allows us to provide quantitative comparisons of MBGP and BGP.

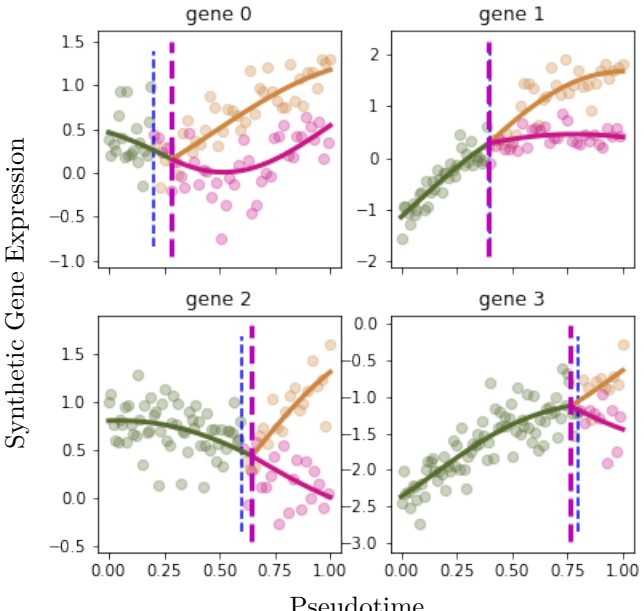

Figure 4: An example of the MBGP model fit on noisy synthetic data with 4 outputs. The data is represented in a scatter plot with inputs $t$ along the $x-$axis and colours indicating branch assignments. The model's mean predictions are the solid lines. The vertical blue dotted lines are the true branching times. The vertical magenta dashed lines are the learned branching times.

### 5.1.1 Setup

We sample noisy data from realisations of MBGP and reject samples where the latent branches cross in more than one location. The MBGP that we sample from has branching points set at target locations distributed equidistantly on $[0, 1]$. Each sample consists of 100 data points. In all experiments that follow we employ the same data point label setup and use the same fitting procedure. This holds true for both MBGP as well as BGP.

We initialise data point label probabilities for latent branches by sampling uniformly from $[0.5, 1]$ for the true label. We then set the data point label prior probabilities to 0.5 for inputs in $[0, 0.8]$ (that is, no prior knowledge) and then 0.8 for the true label for inputs in $[0.8, 1]$, which we will refer to as the "informative prior". Note that even this "informative prior" is weaker than some of the priors explored in the original BGP paper, see §3.4 for details.

Unless explicitly stated otherwise, the fitting procedure is a multiple restart from 4 different branching points, namely, $0.2, 0.4, 0.6, 0.8$ applied uniformly to all outputs. For each restart, we perform gradient descent along the evidence lower bound surface using L-BFGS-B. We demonstrate a typical model fit in Figure 4.

### 5.1.2 MBGP assigns data point labels more accurately

One of the key hypothesised advantages of using a model that considers all outputs at the same time is that by avoiding assignment inconsistencies, namely the biologically impossible situation where cell A is assigned to one branch in gene 0 and a different branch in gene 1, it should improve cell label assignment accuracy as it uses information from all genes.

In order to test the hypothesis we drew 10 random noisy synthetic samples as described above. We then constructed MBGP and BGP models with the ground truth branching times. In contrast to the other

synthetic data experiments we kept the branching times fixed to their ground truth locations and only optimised the other parameters, most notably the data point assignment variational parameters.

We found that MBGP achieved a better mean percentage of correct data point labels at 96.7±0.7% compared to BGP's 92.2±0.5% (the ± value is the standard error). For more detail, see Figure 5. We note that achieving 100% accuracy on data point assignment is a challenging task given the high dataset noise.

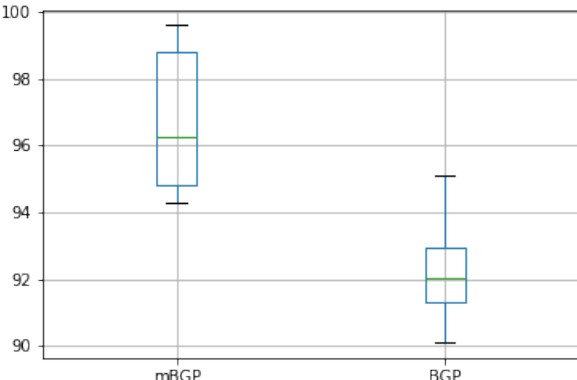

Figure 5: Boxplots of data point assignment correctness averaged over all outputs for each of the 10 samples in MBGP vs BGP. Reported as a percentage, so that 100% is the best score you can hope for. We can see MBGP outperforming BGP and in fact being close to the theoretical highest score.

We also looked at the distribution of incorrect data point labels over inputs. It showed that MBGP performs similarly to BGP at early and late inputs, but holds a considerable advantage in the range of $[0.3, 0.8]$, see Figure 6. The equivalence on $[0.8, 1]$ is not surprising as both models use the informative data point label prior for these inputs. The full per-sample results are reported in the appendix.

Overall, we note MBGP's impressive performance on the data point labeling problem. It suggests that alongside an accurate model fitting procedure for branching points MBGP would constitute a powerful tool for branching time and data point assignment analysis.

### 5.1.3 MBGP improves branching time estimates

The final synthetic data experiment looks at the overall performance of MBGP vs BGP when we're learning both data point assignments and branching locations.

It also explores the inconsistently labelled data point problem in BGP. This happens when a data point is assigned to different branches in e.g. gene 0 and gene 1, which is biologically unrealistic.

We draw 20 random noisy samples and fit MBGP as well as BGP models using the setup described in Section 5.1.1. We measure (i) the root mean square error ($RMSE$) between the learned and ground truth branching points, (ii) the number of correctly learned data point labels, and (iii) the number of inconsistently labeled data points by BGP.

The overall $RMSE$ and correct data point label performance for both models is summarised in Table 2, which shows MBGP outperforming BGP on both measures. We further see in Table 3 that the difference in performance is significant for both $RMSE$ as well as the percentage of correctly learned data point labels.

We note however that the absolute values of $RMSE$ are quite high for MBGP and we hypothesise that this is the key factor in the drop of correct data point label percentage from 96.7% in Section 5.1.2 (where we used the ground truth branching locations) to 86.4% here. It therefore seems that improvements in learning branching locations would be a worthwhile future research direction.

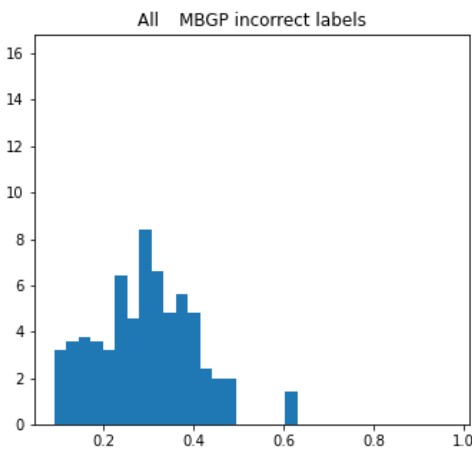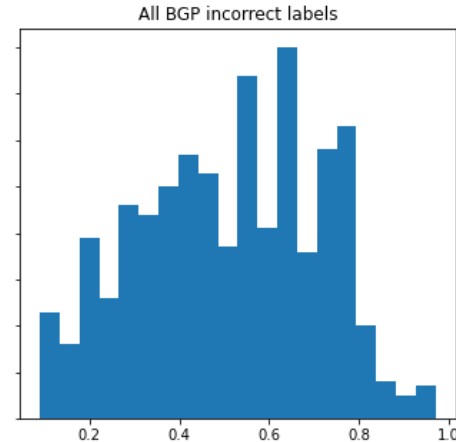

Figure 6: Histogram of all incorrect data point assignments across all samples and all outputs. The $x$-axis is the input of the incorrectly labeled data point. The $y$-axis is normalised to represent the incorrect data points in each bin as a percentage of all data points in that bin. We see that MBGP significantly improves on the data point labeling task and hardly makes any mistakes after 0.5.

Table 2: For each sample, we compute (i) the root mean square error ($RMSE$) between the true and learned branching points (lower is better), and (ii) % - the percentage of correctly learned data point labels (higher is better). We see that in both cases MBGP outperforms BGP, but we note that we see relatively high standard deviations. We believe this is down to the high noise level in the synthetic data.

|  | Mean | Standard deviation |
|---|---|---|
| $RMSE_{MBGP}$ | 0.178 | 0.074 |
| $RMSE_{BGP}$ | 0.222 | 0.070 |
| $\%_{MBGP}$ | 86.4 | 5.1 |
| $\%_{BGP}$ | 80.4 | 4.1 |

Finally, we report that BGP on average has $32.0 \pm 9.8$ inconsistently assigned data points out of a total of 100.

We note that due to the high noise level in random samples, some of them can be extremely difficult to model, see the appendix for some examples.

## 5.2 Hematopoiesis single-cell RNA-seq

We apply the MBGP model on the gene expression of mouse hematopoietic stem cells used in Paul et al. (2015). As in Boukouvalas et al. (2018) we have used the Wishbone algorithm Setty et al. (2016) to derive the pseudotime of each cell as well as the global branching structure. The model is also readily applicable using pseudotime from other methods such as Monocle (Qiu et al., 2017).

Boukouvalas et al. (2018) have given a branching time network of eight genes (Figure 7 (a)) for which they found the highest evidence of branching. In the branching time network shown in Figure 7 (a), the most probable branching time is annotated with each gene and directed edges are used to represent pairwise ordering of genes based on the most likely gene-specific branching time. Boukouvalas et al. (2018) have grouped genes based on the branching time order relationships. For instance, both PRTN3 and CTSG are early branching genes and branch before ELANE, GSTM1, CAR2 and CAR1. On the other hand, MPO and CALR (yellow group) are also early branching genes and branch before GSTM1, CAR2 and CAR1, but

Table 3: For each sample, we compute (i) $\Delta_{RMSE} := RMSE_{BGP} - RMSE_{MBGP}$ where $RMSE$ is the root mean square error between the true and learned branching points, and (ii) $\Delta_\% := \%(MBGP) - \%(BGP)$ where $\%$ denotes the percentage of correctly learned data point labels. In both cases, values above 0 mean that MBGP is outperforming BGP, whereas values below 0 show the opposite. We report the mean and standard error over the 20 samples. As the reader can see, MBGP outperforms BGP.

|  | Mean | Standard error |
|---|---|---|
| $\Delta_{RMSE}$ | 0.0442 | 0.0193 |
| $\Delta_\%$ | 5.98 | 1.40 |

not necessarily before ELANE. Finally, the third group consists of genes ELANE and GSTM1 (blue group) that branch before the gene CAR1. The genes CAR1 and CAR2 have the latest branching time.

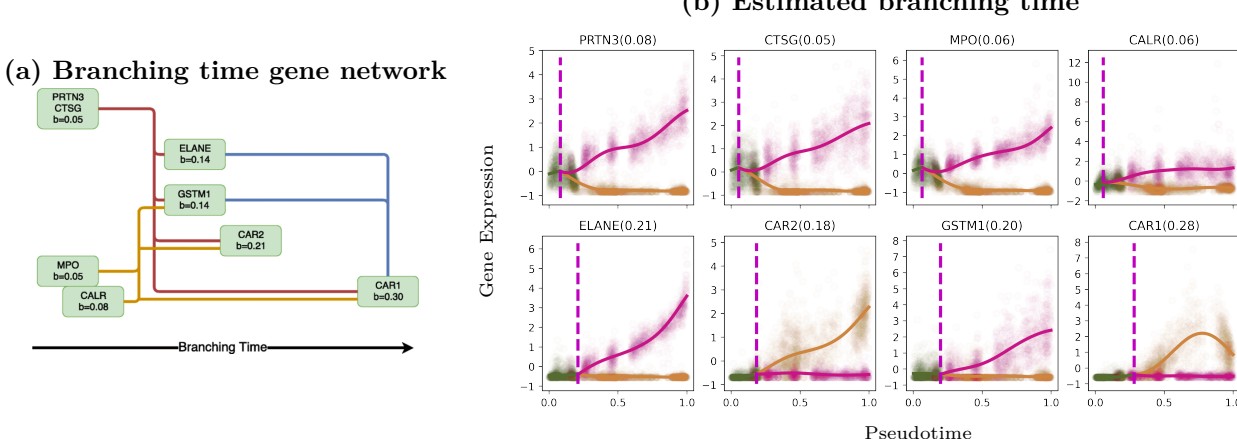

Figure 7: Mouse haematopoietic stem cells (Paul et al., 2015): **(a)**: Estimated gene branching time network from Boukouvalas et al. (2018) : the most probable branching time of each gene is shown. The directed edges are used to represent the pairwise ordering of genes based on their branching time. The edge colours are representing group of genes having the same later branching genes. **(b)**: The MBGP fit. The cells (points) are coloured according to the global branching pattern derived using the Wishbone (Setty et al., 2016) algorithm. The dashed line in each sub-figure represents the estimated branching time for each gene, which is also depicted inside the parenthesis in each sub-figure caption.

Although BGP finds a plausible ordering of the genes for this data, we showed in Table 1 there are large number of inconsistencies of cells to branches when using the original BGP algorithm. Therefore we have applied the MBGP model on the expression profiles of these eight genes which ensures consistency of cell to branch assignments. To speed up inference, we have randomly sub-sampled the data and have used 1000 cells for the inference and have used the sparse model with 60 inducing points.

Figure 7 (b) shows the MBGP model fit on this network of genes. In each sub-figure, the dashed line represents the inferred gene-specific branching time. We reproduce the same pairwise ordering of genes published in Boukouvalas et al. (2018) wherein it was found to be biologically meaningful. If we compare the inferred gene-specific branching times from Figure 7 (b), we see that our model has successfully reproduced the same branching time network of Figure 7 (a).

### 5.2.1 MBGP improves cell assignment accurately

In Appendix C we show examples of cell label inconsistency when applying the BGP algorithm to the mouse haematopoietic stem cell data. In contrast, MBGP ensures by construction no such inconsistencies can occur.

Additional while MBGP ensures the cell assignment consistency, it also improves the posterior cell assignments. We have analysed the posterior cell assignment by MBGP which follows the global branching pattern provided as prior. As the model has learned the gene-specific branching points for a number of genes earlier than the global branching point (main paper Figure 6(b)), therefore, few cells from the trunk state become biased towards either of the branches to facilitate the identification of these pioneer genes. We have summarised this result in Table 4 where a posterior cell assignment confidence of 0.80 is used to determine a cell has been assigned to a branch.

Table 4: The number of cells switches states from prior to posterior as a result of the MBGP model inference for the mouse haematopoietic stem cell data (Paul et al., 2015). A posterior cell assignment confidence of 0.80 is used to determine a cell has been assigned to a branch. The model learns the gene-specific branching points for a number of genes earlier than the global branching point (main paper Figure 6(b)), therefore, few cells from the trunk state become biased towards either of the branches to facilitate the identification of these pioneer genes.

|  | Number of cells in prior | Number of cells in posterior |
|---|---|---|
| Trunk | 278 | 187 |
| Branch 1 | 423 | 466 |
| Branch 2 | 299 | 347 |

## 6 Conclusion

The BGP model was developed to infer branching events from single-cell gene expression data. We uncovered its main limitations, namely inconsistent data point assignments and heavy computational requirements. To mitigate these limitations, we have presented the multivariate BGP (MBGP) model that extends BGP by sharing data point assignments across outputs (genes). On both simulated and mouse hematopoietic stem cell gene expression data we have shown that the MBGP model achieves similar or better branching time accuracy to the original BGP model whilst addressing the assignment inconsistency issue, making model inferences more biologically relevant.

In Section 5.1.2 we showed that when the MBGP model does not have to infer the branching locations, it performs the cell assignment task with impressive accuracy. However, we have also seen in Section 5.1.3 that it does not reach the same accuracy when it also has to learn the branching locations. Thus exploring improvements in the learning of branching locations seems like a valuable future research direction.

Although MBGP is faster than the original BGP formulation, inference is still time-consuming for a large number of outputs since a GP function has to be learned for each output. This also ignores dependencies between gene expression profiles from different genes. Future work could develop a multiple output formulation where individual genes are functional combinations of a smaller set of latent GPs (Wilson et al., 2011). This would allow our approach to scale better to many thousands of outputs, making the approach more practical for large scale single cell branching analysis.

Finally, we note that during the review process of this paper a new method, scFates (Faure et al., 2022), was released that also aims to identify early-branching genes. Direct comparisons between MBGP and scFates are of obvious interest, but have not been evaluated in this paper.

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

## A   Sparse collapsed variational lower bound

We follow similar steps to Lázaro-Gredilla et al. (2012) to derive a sparse variation bound from the full data variational bound derived in Section 4.1 and shown in equation **??**

Starting from the bound in equation **??**, we integrate out the latent GP functions $\boldsymbol{F}$ to derive the collapsed variational bound for the marginal log likelihood.

$$
\begin{aligned}
\log p\left(\boldsymbol{Y}\right) \geq & \\
& - \frac{ND}{2}\log(2\pi\sigma^2) - \frac{1}{2\sigma^2}\boldsymbol{Y}^\top\boldsymbol{Y} \\
& - \frac{D}{2}\sum_{d=1}^{D}\log|\boldsymbol{K}_{ff|b_d}| \\
& - \frac{D}{2}\sum_{d=1}^{D}\log\left|\boldsymbol{A}_d\sigma^{-2} + \boldsymbol{K}_{ff|b_d}^{-1}\right| \\
& + \frac{D}{2}\sum_{d=1}^{D}\sigma^{-4}\boldsymbol{y}_d^\top\boldsymbol{U}_d\left(\boldsymbol{A}_d\sigma^{-2} + \boldsymbol{K}_{ff|b_d}^{-1}\right)^{-1}\boldsymbol{U}_d^\top\boldsymbol{y}_d \\
& - \sum_{d=1}^{D}D_{\mathrm{KL}}\left[Q\left(\boldsymbol{Z}\right)||P\left(\boldsymbol{Z}|b_d\right)\right] \ ,
\end{aligned}
\tag{13}
$$

where $\boldsymbol{K}_{ff|b_d}$ denotes the covariance matrix dependent on the output branching parameter $b_d$.

The sparse variation lower bound for our model $L_s$ is

$$
\begin{aligned}
L_s \triangleq & - \frac{ND}{2}\log(2\pi\sigma^2) - \frac{1}{2\sigma^2}\boldsymbol{Y}^T\boldsymbol{Y} \\
& - \frac{1}{2}\sum_{d=1}^{D}\left[\log|\boldsymbol{P}_d| - \boldsymbol{c}_d^\top\boldsymbol{c}_d\right] \\
& - \frac{1}{2\sigma^2}\sum_{d=1}^{D}\left[\mathrm{tr}\left(\boldsymbol{A}_d\boldsymbol{K}_{ff_d}\right) - \mathrm{tr}\left(\boldsymbol{A}_d\boldsymbol{K}_{fu_d}\boldsymbol{K}_{uu_d}^{-1}\boldsymbol{K}_{uf_d}\right)\right] \\
& - \sum_{d=1}^{D}D_{D_{\mathrm{KL}}}\left[Q\left(\boldsymbol{Z}\right)||P\left(\boldsymbol{Z}|b_d\right)\right] \ ,
\end{aligned}
\tag{14}
$$

where we have defined

$$
\begin{aligned}
\boldsymbol{P}_d &\triangleq I + \boldsymbol{L}_d^{-1}\boldsymbol{K}_{sf|b_d}\boldsymbol{A}_d\boldsymbol{K}_{fs|b_d}\boldsymbol{L}_d^{-\top}\sigma^{-2} \ , \\
\boldsymbol{c}_d &\triangleq \boldsymbol{R}_d^{-1}\boldsymbol{L}_d^{-1}\boldsymbol{K}_{sf|b_d}\boldsymbol{U}_d^\top\boldsymbol{Y}_d\sigma^{-2} \ ,
\end{aligned}
$$

and $s$ is the set of inducing points and the Choleksy factors $\boldsymbol{L}_d$ and $\boldsymbol{R}_d$ are defined as

$$
\begin{aligned}
\boldsymbol{K}_{uu_d} &\triangleq \boldsymbol{L}_d\boldsymbol{L}_d^\top \ , \\
\boldsymbol{P}_d &\triangleq \boldsymbol{R}_d\boldsymbol{R}_d^\top \ .
\end{aligned}
$$

## B  Detailed experimental results on synthetic data

### B.1  Labelling performance

The full per-sample results are reported in Table 5. We have already reported the key summary statistics in the main part of the paper, so we omit further discussion. We also show in Figure 8 an example of the MBGP model fit on the noisy synthetic data. We note again MBGP's impressive performance on the data point labeling.

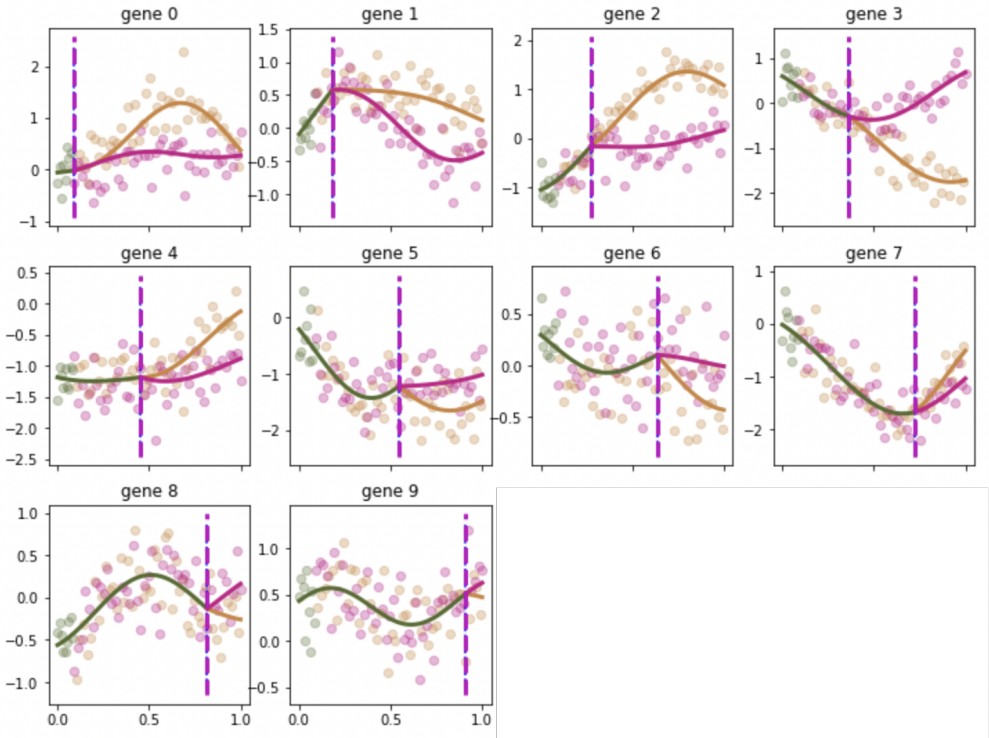

Figure 8: An example of the MBGP model fit on noisy synthetic data with 10 outputs. The data is represented in a scatter plot with input along the $x$ axis and colours indicating branch assignments. The model's mean predictions are the solid lines. The vertical magenta dashed lines are the true branching times - we do not learn branching locations in this example.
We draw the reader's attention to the data points around the true branching point and suggest that it is a highly nontrivial problem to assign them to branches given the noise levels.

Table 5: This table contains the per-sample results for MBGP and BGP. Each row corresponds to a different 10 output noisy synthetic dataset. $inc_{BGP}$ only applies to the BGP model and describes how many data points have inconsistent assignments across the outputs, that is, for how many data points do we assign them to one branch in output A and to a different branch in output B? We note that lower is better and 100 is the absolute worst result you could get while 0 is the best result. % is the percentage of correctly learned data point labels (higher is better). In this measure, we ignore inconsistent data point assignments and simply average over all outputs.

We can see that the MBGP model is able to infer data point assignment with higher accuracy, although both models are able to achieve accuracy $> 90\%$. We suspect MBGP outperforms BGP due to its multivariate nature allowing it to more accurately learn data point assignments around branching locations where there is significant uncertainty.

|   | $inc_{BGP}$ | $\%(MBGP)$ | $\%(BGP)$ |
|---|---|---|---|
| 0 | 22 | 95.3 | 91.3 |
| 1 | 24 | 94.4 | 91.5 |
| 2 | 18 | 99.6 | 93.8 |
| 3 | 26 | 95.2 | 91.3 |
| 4 | 23 | 94.7 | 90.3 |
| 5 | 19 | 97.6 | 93.0 |
| 6 | 18 | 97.2 | 92.7 |
| 7 | 16 | 99.3 | 95.1 |
| 8 | 27 | 99.2 | 92.6 |
| 9 | 31 | 94.3 | 90.1 |

### B.2 Details on synthetic data branching performance

We provide more details on the synthetic data experiments discussed in Section 5.1.3. We remind the reader that the final synthetic data experiment looks at the overall performance of MBGP vs BGP when we're learning both data point assignments and branching locations.

The per-sample $RMSE$ and correct data point label performance for both models is provided in Table 6. We have already reported the key summary statistics in the main part of the paper, so we omit further discussion. However, we will now look at two different regimes: "easy" and "hard" to infer branching data, corresponding to samples 2 and 14 respectively in Table 6.

The model fits on sample 2 from Table 6 are discussed in Figure 9 and Figure 10, for MBGP and BGP respectively. See the captions for a discussion.

The model fits on sample 14 from Table 6 are displayed in Figure 11, which pertains to MBGP, and Figure 12, which pertains to BGP. See the captions for a discussion.

Finally, for completeness, we also provide (i) the distribution of incorrect data point labels across input, see Figure 13, and (ii) the mean percentage of correct data point labels, see Figure 14.

## C Mouse haematopoietic stem cells: Inconsistent cell assignment by BGP

In the BGP model, the inference is performed independently per gene which causes inconsistent cell assignments across different genes. To minimise cell assignment inconsistency we use a very strong cell assignment prior probability of 0.95 as suggested by Boukouvalas et al. (2018), but the problems remains to a great extent. Figure 15 shows the expression profiles as well as BGP fits of eight genes where triangle markers are used to indicate the cells that have been assigned to different branches inconsistently. Among the 1000 cells used in the inference of BGP, 468 cells show inconsistency across all eight genes. If we would have included more genes this number would have been greater. Thus the cell assignment reported by the BGP is not informative and likely to be misleading.

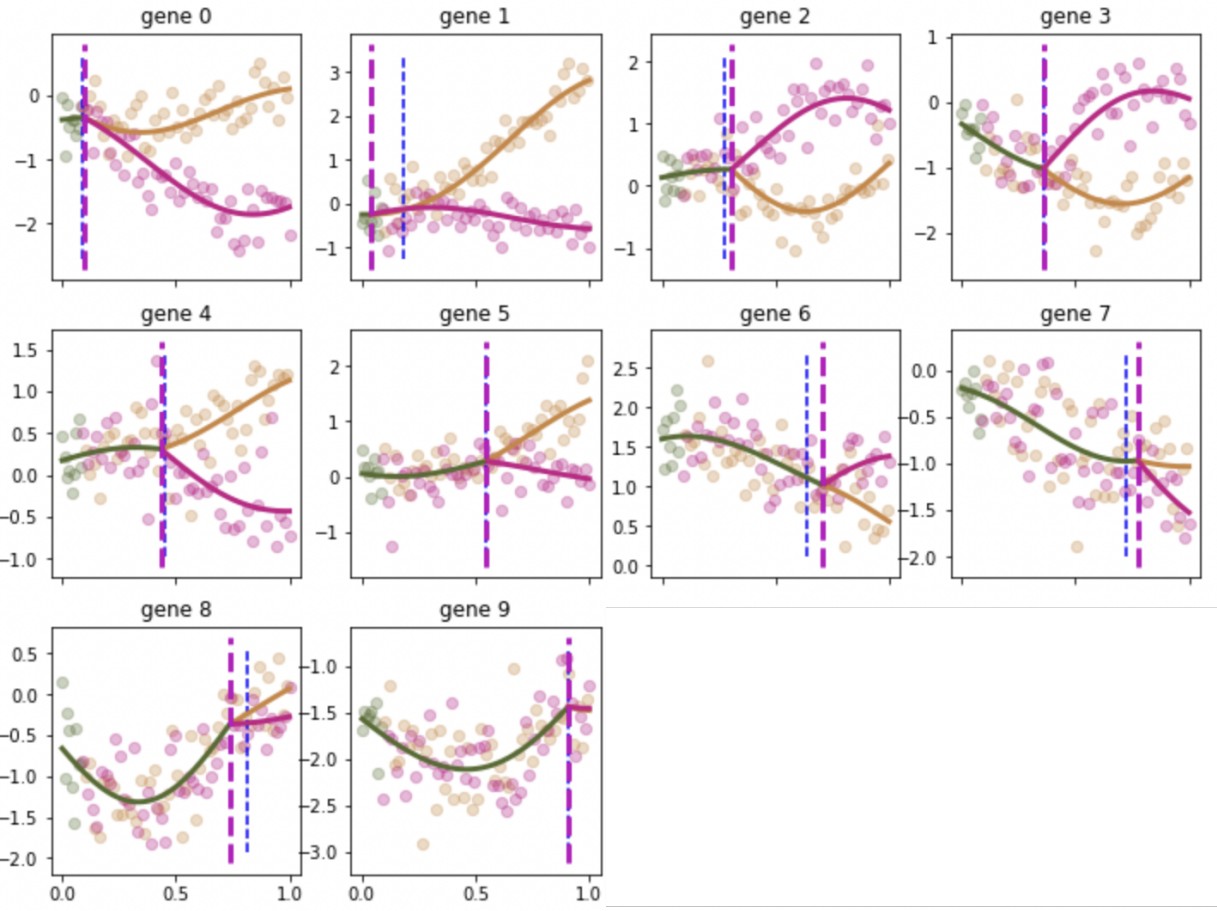

Figure 9: This Figure shows the MBGP model fit on sample 2 from Table 6. The data is represented in a scatter plot with inputs along the $x$ axis and colours indicating branch assignments. The model's mean predictions are the solid lines. The vertical blue dotted lines are the true branching times. The vertical magenta dashed lines are the learned branching times.

Overall the model does a good job of learning the true branching times and thus does well on data point assignment: $\%_{MBGP} = 93.7\%$ vs $\%_{BGP} = 81.3\%$, see Table 6. We hypothesise that this could be due to the early branching outputs, namely, output 0 and output 1 providing useful evidence for data point assignment that the other outputs then benefit from. Note, in particular, how MBGP learns the correct branching time for output 7 whereas BGP learns incorrect dynamics, see Figure 10.

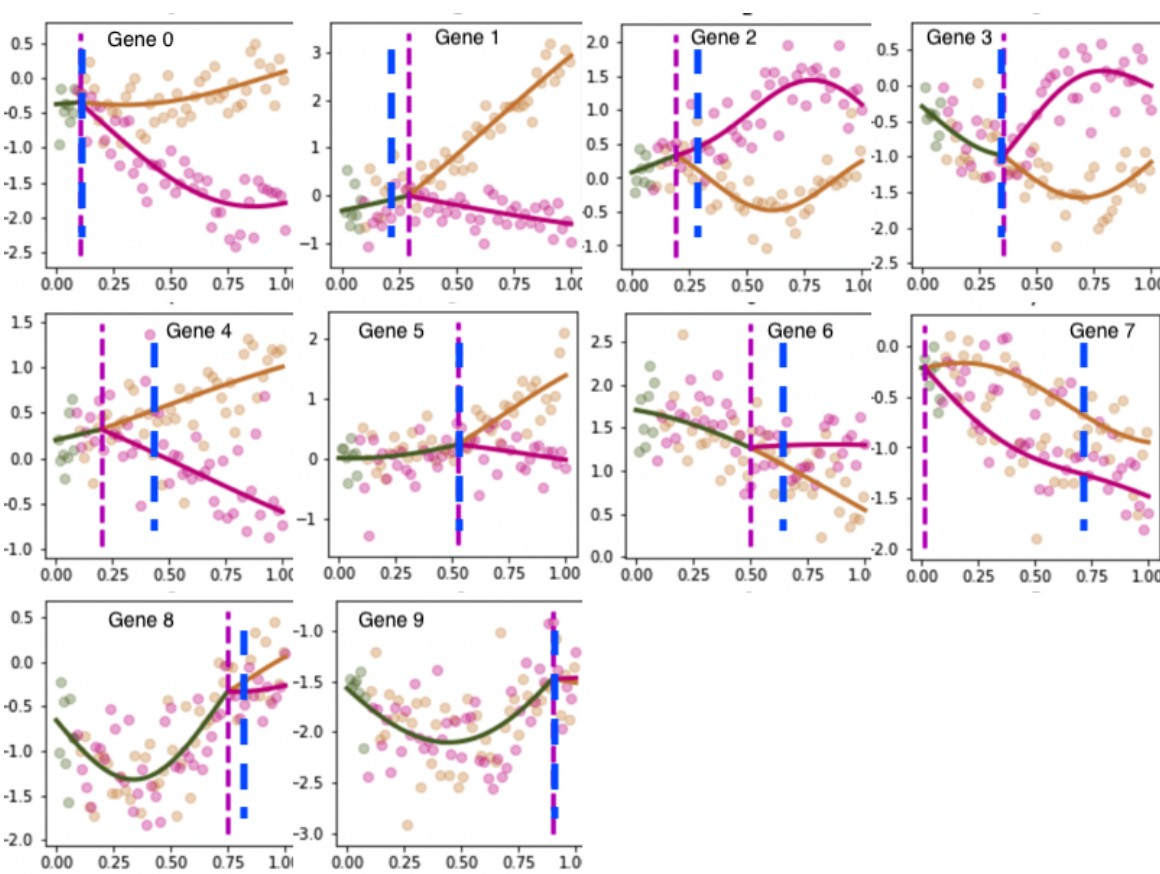

Figure 10: This Figure shows the BGP model fit on sample 2 from Table 6. The data is represented in a scatter plot with input along the $x$ axis and colours indicating branch assignments. The model's mean predictions are the solid lines. The vertical blue dashed lines are the true branching times. The vertical magenta dashed lines are the learned branching times.
See Figure 9 for the key discussion points.

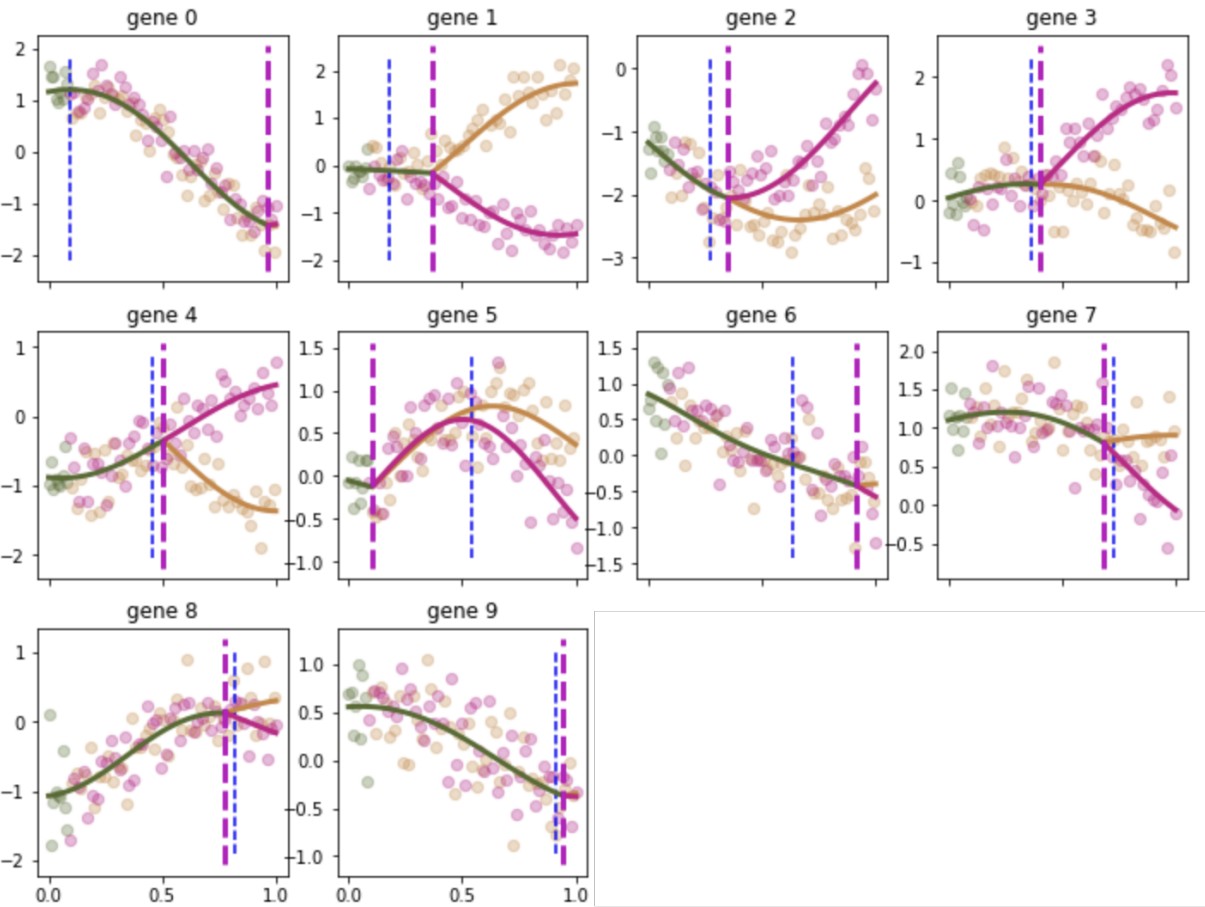

Figure 11: This Figure shows the MBGP model fit on sample 14 from Table 6. The data is represented in a scatter plot with input along the $x$ axis and colours indicating branch assignments. The model's mean predictions are the solid lines. The vertical blue dotted lines are the true branching times. The vertical magenta dashed lines are the learned branching times.

Overall the model struggles with learning the true branching times and thus data point assignment suffers as well: $\%_{MBGP} = 79.4\%$ vs $\%_{BGP} = 79.8\%$, see Table 6. We hypothesise that this could be due to the early branching output 0 being hard to model due to the branches staying close to each other. In particular, this means that output 0 contributes incorrect data point assignment suggestions to the other outputs thus making the model's task harder than it should be.

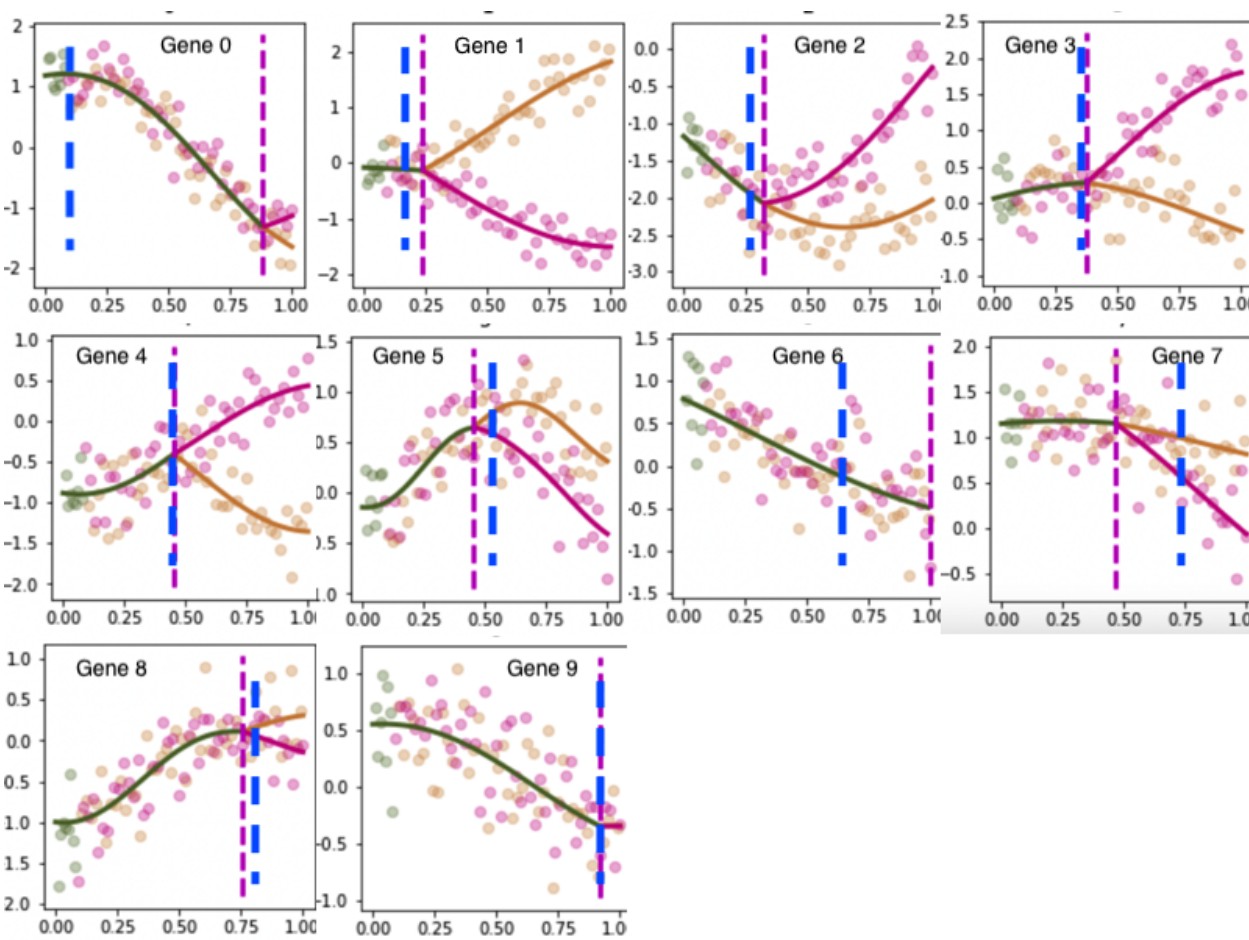

Figure 12: This Figure shows the BGP model fit on sample 14 from Table 6. The data is represented in a scatter plot with input along the $x$ axis and colours indicating branch assignments. The model's mean predictions are the solid lines. The vertical blue dashed lines are the true branching times. The vertical magenta dashed lines are the learned branching times.
See Figure 11 for the key discussion points.

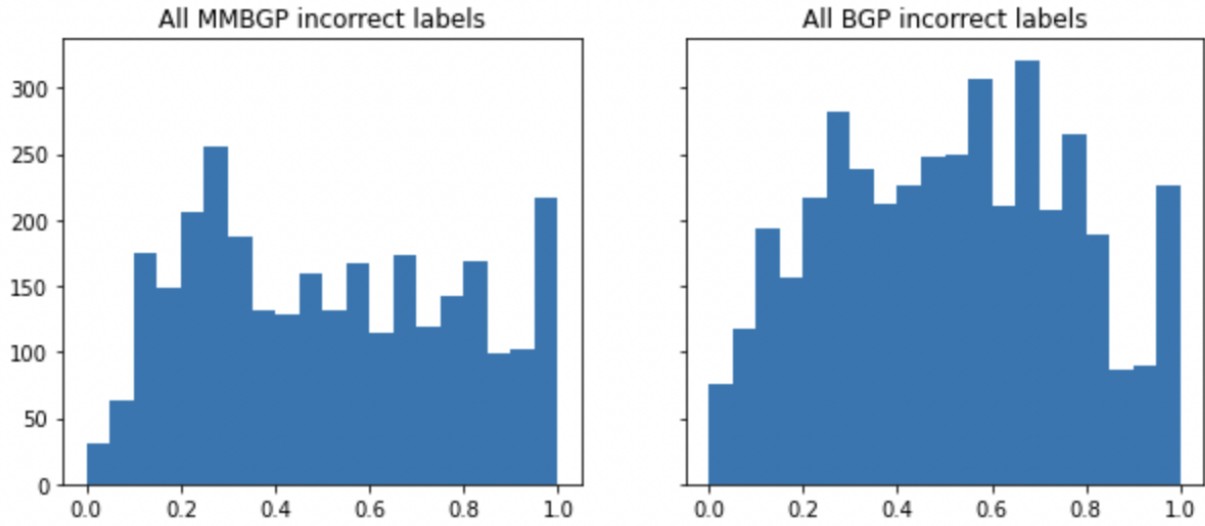

Figure 13: Histogram of all incorrect data point assignments across all samples and all outputs. The $x$-axis is the input of the incorrectly labeled data point. The $y$-axis is normalised to represent the incorrect data points in each bin as a percentage of all data points in that bin.

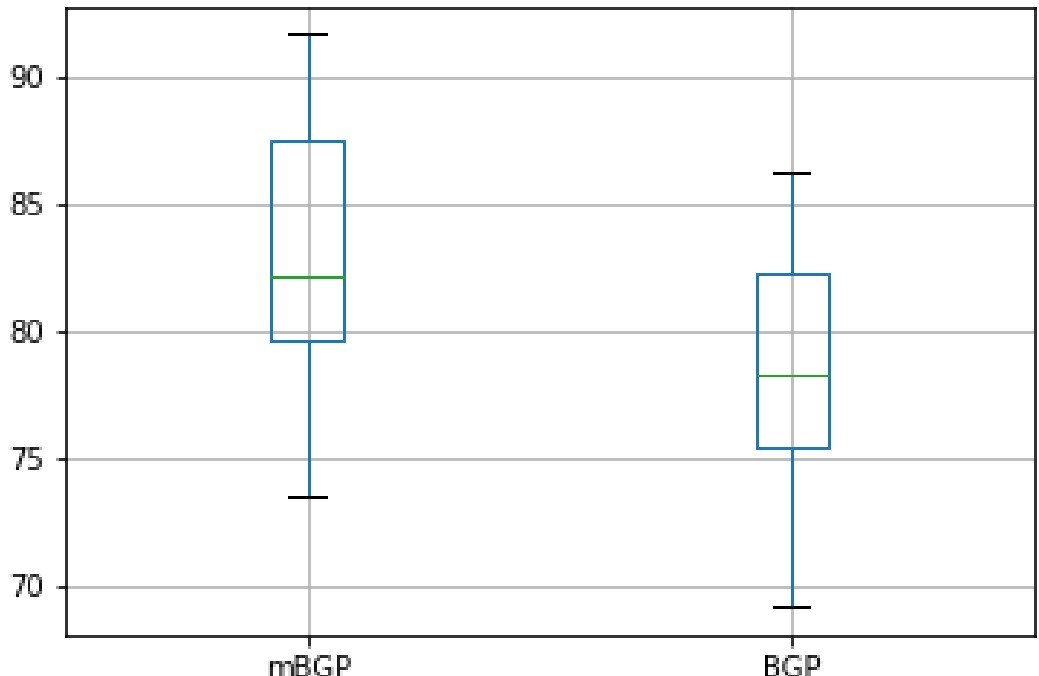

Figure 14: Boxplots of data point assignment correctness averaged over all outputs for each of the 20 samples in MBGP vs BGP. Reported as a percentage, so that 100% is the best score you can hope for.

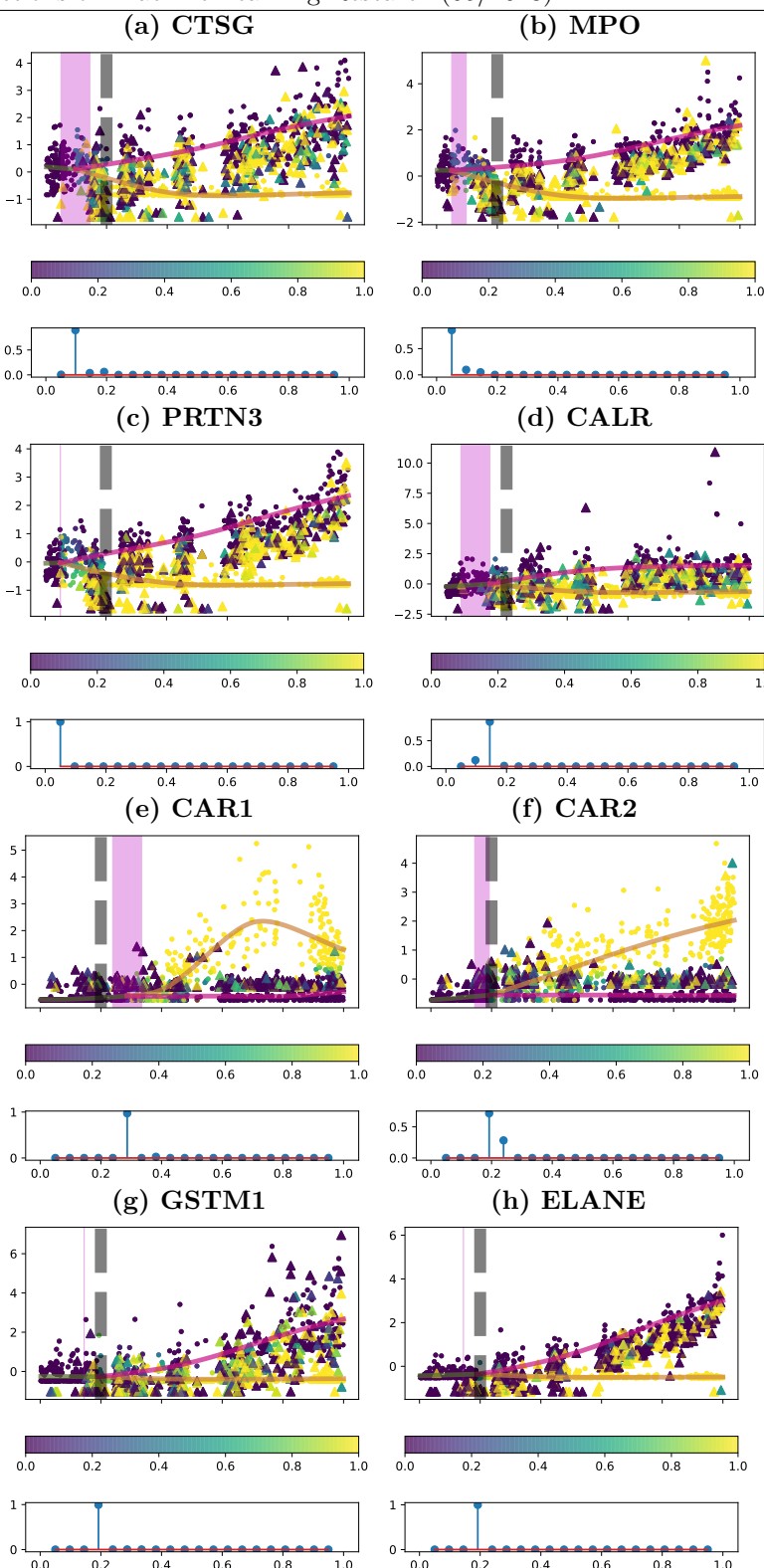

Figure 15: Mouse haematopoietic stem cells (Paul et al., 2015): Inconsistent cell assignment across the top eight branching genes. The 1000 randomly sub-sampled cells that have been used for inference are coloured according to the posterior global branching pattern derived using the BGP algorithm. The points are used to represent the cells have been assigned to the same branches, i.e. consistent across the eight genes. On the other hand, the triangles represent the inconsistent cell assignment, i.e. cells that have been assigned to different branches across the eight genes.

Table 6: This table contains the per-sample results for MBGP and BGP. Each row corresponds to a different 10 output noisy synthetic dataset. The $RMSE$ columns measure the root mean square error between the learned and ground truth branching points. $inc_{BGP}$ only applies to the BGP model and describes how many data points have inconsistent assignments across the outputs, that is, for how many data points do we assign them to one branch in output A and to a different branch in output B? We note that lower is better and 100 is the absolute worst result you could get while 0 is the best result. % is the percentage of correctly learned data point labels (higher is better). In this measure, we ignore inconsistent assignments and simply average over all outputs. Finally, $\Delta_\% := \%(MBGP) - \%(BGP)$; above 0 means MBGP is learning data point assignment better, below 0 means BGP is doing better.

As reported in the main body of the paper, MBGP outperforms BGP. In addition to that we highlight that there's a correlation between how well MBGP learns the branching points and how well it assigns data point labels. When it learns the branching point locations better or equal to BGP, then it seems to consistently do better at data point assignment. As before, this motivates exploring schemes for learning more accurate branching locations.

| | $RMSE_{MBGP}$ | $RMSE_{BGP}$ | $inc_{BGP}$ | $\%(MBGP)$ | $\%(BGP)$ | $\Delta_\%$ |
|---|---|---|---|---|---|---|
| 0 | 0.155656 | 0.161103 | 45 | 87.9 | 80.5 | 7.4 |
| 1 | 0.107528 | 0.172943 | 43 | 91.2 | 83.9 | 7.3 |
| 2 | 0.060403 | 0.248472 | 47 | 93.7 | 81.3 | 12.4 |
| 3 | 0.144460 | 0.209304 | 42 | 90.1 | 79.8 | 10.3 |
| 4 | 0.107687 | 0.122775 | 32 | 92.5 | 83.5 | 9.0 |
| 5 | 0.231522 | 0.171707 | 23 | 82.3 | 85.2 | -2.9 |
| 6 | 0.145352 | 0.301008 | 43 | 87.7 | 75.0 | 12.7 |
| 7 | 0.125235 | 0.335639 | 34 | 89.2 | 77.8 | 11.4 |
| 8 | 0.277916 | 0.280737 | 27 | 80.4 | 78.1 | 2.3 |
| 9 | 0.149851 | 0.274390 | 37 | 89.7 | 76.9 | 12.8 |
| 10 | 0.249903 | 0.329334 | 41 | 81.2 | 69.4 | 11.8 |
| 11 | 0.239118 | 0.212270 | 25 | 79.8 | 81.2 | -1.4 |
| 12 | 0.172359 | 0.123819 | 26 | 85.2 | 85.6 | -0.4 |
| 13 | 0.149704 | 0.187880 | 13 | 87.1 | 85.0 | 2.1 |
| 14 | 0.329045 | 0.290695 | 20 | 79.4 | 79.8 | -0.4 |
| 15 | 0.210808 | 0.230580 | 29 | 86.6 | 77.7 | 8.9 |
| 16 | 0.116043 | 0.114808 | 40 | 91.9 | 83.2 | 8.7 |
| 17 | 0.138261 | 0.296907 | 21 | 89.2 | 77.3 | 11.9 |
| 18 | 0.134857 | 0.165543 | 27 | 88.1 | 83.6 | 4.5 |
| 19 | 0.318348 | 0.218915 | 25 | 74.8 | 83.6 | -8.8 |

Figure 16 shows an example of the BGP model applied on an early branching gene from a mouse hematopoietic stem cell dataset (Paul et al., 2015). The data contain 4423 cells. The log normalised data were transformed to zero mean for each individual gene. The DDRTree algorithm from Monocle 2 (Qiu et al., 2016) is used to infer the global cellular branching pattern as well as pseudotime for each cell. We have used a sub-sample of 870 cells and 30 inducing points to speed up the model inference. Figure 16 (a) shows the prior cell assignment of all 4423 cells which has been derived by running the Monocle 2 algorithm. This global branching structure is used as the informative prior. Figure 16 (b) shows the posterior cell assignment of 870 cells that have been used in the inference (top sub-panel). The cells that are away from the global branching time (the black dashed line in Figure 16 (a)) have been assigned to either of the branches with high confidence. On the other hand, the cells closer to or around the global branching point possess high level of uncertainty. This is also the case for the cells equidistant from both branches. The bottom sub-panel of Figure 16 (b) shows the inferred gene-specific branching dynamics along with the posterior probability of the branching location. This uncertainty has been also reflected in Figure 16 (a) where the magenta background depicts the uncertainty associated with the gene-specific branching time (the blue solid line). The uncertain regions shown in both figures (magenta background in Figure 16 (a) and green dots near branching-time in Figure 16 (b)) are indicative of how cell assignment uncertainty is incorporated into the gene-specific

branching-time posterior uncertainty. It emphasises one of the major benefits of developing probabilistic methods like the BGP for downstream analysis. As single-cell data are noisy, the cell assignment to different branches should be probabilistic while identifying the gene-specific branching dynamics.

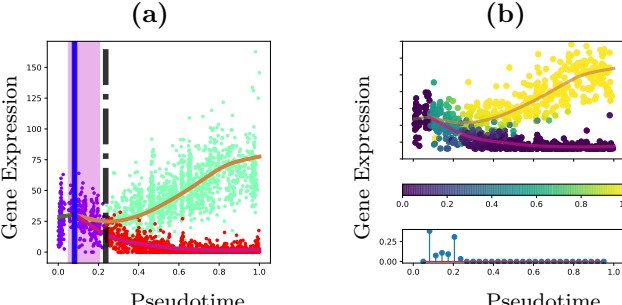

Figure 16: Mouse haematopoietic stem cells (Paul et al., 2015): BGP fit for the early branching gene MPO. The DDRTree algorithm of Monocle 2 is used to estimate the pseudotime for each cell as well as the global branching pattern. **(a)**: Monocle 2 global assignment of cells to the trunk state (purple) and two branches (green and red). The black dashed line is the global branching time. The most probable gene branching time is shown by the blue solid line along with posterior uncertainty over the branching location (magenta background). **(b)**: The top sub-panel shows the posterior cell assignment uncertainty for the cells used in the inference. Cell assignment to one of the branches is depicted. The bottom sub-panel represents the posterior probability distribution over the gene-specific branching location.

