# OpenReview forum: "Modelling sequential branching dynamics with a multivariate branching Gaussian process"
_TMLR — Accepted by TMLR_

### Review · Reviewer_TGmh · 2022-12-19

**Summary Of Contributions:**

The Overlapping Mixture of Gaussian Processes (OMGP) and its variants like the Branching Gaussian Process (BGP) are models designed to detect bifurcation points in single-cell gene expression data. However, the current BGP model has a limitation in that it independently infers the assignment of observations to latent functions for each output dimension (gene), which can lead to inconsistent assignments and reduced accuracy in branching time inference. In this paper, the authors propose a new model called the multivariate branching Gaussian process (MBGP) and perform joint branch assignment inference across multiple output dimensions to ensure assignment consistency and improve accuracy in branching time inference and assignment accuracy. The authors test the effectiveness of their so-called “multivariate branching Gaussian process” model on synthetic data and a real dataset of mouse stem cells and develop a variational inference procedure to scale the model to large datasets.

**Audience:**

Yes

**Broader Impact Concerns:**

No Broader Impact Statement section is included. However, in my opinion, there are no obvious concerns regarding the ethical implications of the work that necessitates the inclusion of this section.

**Claims And Evidence:**

Yes

**Requested Changes:**

Broadly speaking, I find the figures in this to be quite difficult to parse. Beyond simple things like adding axis labels (which are missing in almost all figures), there is generally some room for improvement in things like figure captions and legends and beyond.

For instance, it isn't sufficient to label subfigures "MPO" and "CTSG" (in Figure 2) even if it is stated explicitly somewhere in the text that these are genes, since this likely won’t immediately occur to the target audience of TMLR.

Figure 2 is tough to parse. Not only are there many different components (markers with different colours and shapes, colour bars, vertical shaded regions, curves, etc.) there are no labels in the form of figure legends, let alone axis labels. Furthermore, it is not possible to understand the figure from the caption alone, one really has to parse through the main text.

### Misc

- On Page 3, a section reference did not render correctly
- Page 9: "Finally, obtain a sparse collapsed variational bound by [...]"

**Strengths And Weaknesses:**

**Significance.** The proposed method (combination of model and inference procedure) seems very bespoke and application-specific. I am unsure if the framework would readily transfer beyond the analysis of gene expression data. Now, within this particular application, I am not well-versed in the domain, so I cannot assess with any confidence the importance or difficulty of solving these particular subproblems.

**Novelty / originality.** The originality of the contribution is nontrivial but modest. Augmenting a model and its inference by going beyond the mean-field assumption is a conceptually simple improvement. Proposing a computationally feasible inference procedure for this model doesn’t appear to be too challenging since the necessary components are already in place. However, combining them into one coherent framework turns out as usual to be not so trivial and this work seems to have done that successfully, as evidenced by their qualitative and quantitative experimental results.

**Clarity / Presentation.** The structure of this paper is well-organized and the writing is mostly clear and effective.

Section 2 defines provides a formal definition of the two interdependent problems that this paper seeks to solve. Section 3 formalizes the existing BGP model and relates it to OMGP. Thereafter, they illustrate the failure modes of this class of models in solving the second subproblem of interest — namely, the cell labelling problem. In brief, it is possible for the same cell to be assigned to varying branches across different genes, which is biologically impossible. Section 4 summarizes their core contribution and Section 5 provides empirical results to validate their method.

In general, the presentation could be adapted more to the ML research community that primarily constitutes the target audience of TMLR. For example, I think there is room for improvement in the problem definition section, which is perhaps a bit too terse and assumes a certain degree of familiarity with gene expression data. Some other issues are also referred to in the Requested Changes section.

**Quality / soundness.** I checked the equations and derivations, particularly those that describe the proposed, and these seem sound and correct. Some equations seem unnecessarily verbose, and could probably be deferred to the appendix. For example eq. 11 explicitly spells out a multivariate normal log density, eq. 12 is quite fleshed out. In terms of the proposed inference procedure, the authors propose a two-stage approach to estimating the GP hyperparameters and the branching points and cell assignments. While this is somewhat justified by intuition, it would be more rigorous to include an ablation study that compares against a joint treatment of these (hyper)parameters. Otherwise, the experimental setup appears to be sound and the empirical improvements over BGP appear to be significant.

---

### Review · Reviewer_zLb8 · 2022-12-23

**Summary Of Contributions:**

Modeling sequential branching dynamics is an interesting topic, especially, it is important for analyzing single-cell gene expression data. In this paper, the author(s) tried to fix the limitations in BGP model and proposed a multivariate BGP version, called MBGP, to jointly analyze branch assignment inference. And through the experiments on synthetic data and a real-world single-cell RNA-seq data, the author(s) demonstrated the advantages of the proposed method.




**Audience:**

Yes

**Broader Impact Concerns:**

No.

**Claims And Evidence:**

No

**Requested Changes:**

Detailedly, I have the following comments/questions:

1. By experiments with only two datasets (that is, one simulated dat set and one real dataset), then it concluded that the proposed method performed better than other existing method(s), I think it is not convincing. Moreover, the author(s) did not provide a comparison of the proposed method with existing method(s) on that real dataset in the main text.

2. In the introduction, the author(s) mentioned several existing methods, why not compare the proposed method with these methods? In experiments, the author(s) should at lease explain the reason.

3.The existing work on which this study is based does not seem too new or current.

After looking at the references of this paper, it seems that only two citations are from 2020 onwards, namely 2020 and 2021. Since there is not a lot of overlap between my research and the work of this paper, I'm not sure whether the author(s) haven't done enough search on the literature, or this field has stagnated. Could the author(s) please explain that?

4. Many issues/typos:

(1) On page 3, Line 5, "??", what does it denote?

(2) In the main text, sometimes mathematical formulas have punctuation at the end and sometimes they don't.

(3) Giving the full name the first time "BGP" appears in the main text, not until section 3.

(4) The caption of Table 1 should end with punctuation.

(5) Please unify the table styles in the main text.

(6) In Figure 5, the numbers on the axes are too small.

(7) In Figure 5, what does MMBGP mean?

(8) Many formatting inconsistencies in the references, please check and correct them carefully.


I look forward to the response from the author(s). Thanks!






**Strengths And Weaknesses:**

## Strong points:

1) The proposed method considered joint inferences across all the genes.

2) MBGP improved accuracy in branching time inference and assignment accuracy, and it is faster than the original BGP.

## Weak points:

1) The empirical experiments and comparisons are not sufficient;

2) The existing work on which this study is based does not seem too new or current;

3) Too many typos and format issues.

In addition, although there are empirical experiments in this paper, I did not find codes related to these experiments, so I am not sure the empirical description/experiments would be enough to reproduce.

---

### Review · Reviewer_NqTz · 2023-02-02

**Summary Of Contributions:**

this paper proposes a Multivariate Branching Gaussian Process, an improvement over BGP, by considering multiple genres at the same time to achieve  label consistency and with better accuracy/efficiency shown. The main innovations are to use shared parameters (Z, H) and direct modeling of the branching time.

**Audience:**

Yes

**Claims And Evidence:**

Yes

**Requested Changes:**


presentation improvement and more empirical evaluations

**Strengths And Weaknesses:**

Strength:
1. the application of cell gene modeling is quite interesting
2. utilization of multi-task
3. clear improved performance

Weakness:
1. only compared against BGP, with many potentially simple multi-task baselines not considered.
2. presentation needs improvements, in particular the early part of the paper.


Other comments:
-The introduction is not very clear in describing the problem, before discussing issues with current problems. What is the input/output of the problem, and what is exactly trajectories? Maybe an illustrative figure here would be helpful. In addition, discussion on why such a problem is important is needed.
-  " global branches" what does this mean? if you have more genes, do they also belong to one of two branches?
- notations are not consistent with each other, and the organization and presentaion is not quite clear: for example, just in problem definition
1. both t and b are used to indicate time.
2. bifurcation branch label should be clearly labeled with a letter and clarify it is an output
3.  a property "consistent cell labelling across genes" shown in the problem definition before even defying the problem is strange.

Other presentation issues:
1. "as we will see in §?? this mathematical convenience comes at a cost to modelling capabilities".
2. highlight the best performing methods in tables

- "are optimised along with other hyper-parameters" generally only parameters are optimized.

---

### Decision · Action_Editors · 2023-04-15

**Recommendation:** Accept with minor revision

**Comment:**

The claims of the paper relate to an improvement of the BGP model and application to single-cell transcription experiments. This claim seems to be sufficiently interesting to the audience to warrant consideration for this venue. Some reviewers commented on lack of comparison to other competing methods and presentation issues that would improve the paper substantially. The authors responded to those comments and offered to update the paper with four specific changes including a more current review of literature. I encourage to make the changes that they suggested in response to the comments.

**Audience:**

The paper is of interest to researchers that are analyzing single-cell transcription data and researchers in Gaussian process models.

**Claims And Evidence:**

The paper introduces a modification of the branching Gaussian process model that incorporates correlation between genes in the branching process. The authors claim this multivariate branching Gaussian process (MGP) model has higher fidelity to biological reality and reduced the frequency of contradictory assignments. The authors summarize the key challenge in their paper, "...a cell may be assigned to branch A for one gene and to branch B for another gene, which makes little biological sense." And they go on to claim, "it is likely to be beneficial to leverage data from multiple genes to consistently solve the cell labelling problem, since there will be high uncertainty given data from only one gene." They evidence this difficulty in Table 1. Section 5.1.2 provides evidence that the MBGP model improves upon BGP in terms of cell labels and section 5.1.3 shows that MBGP improves upon BGP in terms of branching time estimates. They demonstrate their method on a small real data example, but do not provide biological validation of the inferences on real data which seems outside of the scope for this paper.